# Validation and limitations of the distress thermometer in identifying depression among metastatic breast cancer patients in Nigeria: Methodological challenges in depression-specific screening validation

Tejen Chavda[1], Abdulrazzaq Lawal[2], Anthonia Sowunmi[3,4], Bolanle Adegboyega[3,4], Adewumi Alabi[3,4], Chen-Chih Chung[5,6], Sylvia Shirima[7], Donaldson F. Conserve[8], Oluwaseun Adebayo Bamodu[7,9]*

1 Milken Institute School of Public Health, George Washington University, Washington, District of Columbia, United States of America, 2 Department of Surgery, College of Medicine, University of Lagos, Lagos, Nigeria, 3 Department of Radiation Biology, Radiotherapy and Radiodiagnosis, College of Medicine, University of Lagos, Lagos, Nigeria, 4 NSIA-LUTH Cancer Centre, Lagos University Teaching Hospital, Idi-Araba, Lagos, Nigeria, 5 Department of Neurology, Taipei Medical University—Shuang Ho Hospital, New Taipei City, Taiwan, 6 Department of Neurology, School of Medicine, College of Medicine, Taipei Medical University, Taipei City, Taiwan, 7 Building Research and Implementation to Drive Growth and Equity in Africa (BRIDGEAfrica), Dar Es Salaam, Tanzania, 8 Department of Community Health and Prevention, Milken Institute School of Public Health, George Washington University, Washington, District of Columbia, United States of America, 9 Bethesda Achievers Consultancy, LLC, Hagerstown, Maryland, United States of America

* dr_bamodu@yahoo.com

## Abstract

### Background

Psychological distress is highly prevalent among patients with metastatic breast cancer (MBC), yet validated screening tools remain scarce in low- and middle-income countries (LMICs). The Distress Thermometer (DT), a globally endorsed screening instrument, lacks robust validation in advanced-stage cancer populations within LMIC contexts, where cultural, linguistic, and healthcare system factors may significantly influence its diagnostic performance.

### Objective

To assess the diagnostic validity of the Distress Thermometer in identifying clinically significant depressive symptoms among Nigerian women with metastatic breast cancer, using the Beck Depression Inventory-II (BDI-II) as the reference standard.

### Design

Cross-sectional diagnostic validation study employing receiver operating characteristic (ROC) analysis to evaluate DT performance, with optimal cutoff identified using Youden's Index.

**Data availability statement:** De-identified participant data may be made available upon reasonable request and approval by the institutional ethics committees, subject to data sharing agreements that ensure participant privacy and appropriate use of data. The de-identified dataset cannot be publicly deposited due to ethical and legal restrictions. Our Institutional Review Board (Lagos University Teaching Hospital Health Research Ethics Committee, IRB ADM/DCST/HREC/APP/7058) explicitly prohibits unrestricted public data sharing for this vulnerable population (women with terminal metastatic breast cancer), as this would violate Nigerian data protection regulations and could compromise patient privacy through indirect identification despite de-identification. Public deposition would violate our IRB-approved protocol and participant consent, which specified controlled data sharing only as imposed by the Lagos University Teaching Hospital Health Research Ethics Committee (LUTH-HREC). However, the complete de-identified dataset is available to qualified researchers upon reasonable request through a controlled access mechanism. This study represents a secondary analysis of de-identified data from the primary study "Access to psychological support, psycho-oncology and psychotherapy among cancer patients in Nigeria" (Principal Investigator: Dr. A.O. Alabi). Data are available from the LUTH-HREC Data Access Committee for researchers who meet criteria for access to confidential data. Institutional Contact for Data Access: Lagos University Teaching Hospital Health Research Ethics Committee (LUTH-HREC) Idi-Araba, Lagos, Nigeria Email: cmd@lasuth.org; info@luth.gov.ng Phone: +234 812 836 4824 IRB Approval Number: ADM/DCST/HREC/APP/7058 Data access requests will be reviewed by the LUTH-HREC to ensure compliance with Nigerian research ethics regulations and patient privacy protections. Qualified researchers must submit: (1) institutional affiliation and qualifications, (2) intended use and analysis plan, and (3) formal data sharing agreement. Approved requests typically processed within 2-4 weeks, with data transfer within 6 months. This controlled access approach complies with Nigerian research ethics regulations, protects our vulnerable patient

## Setting/participants

A total of 309 histologically confirmed de-identified metastatic breast cancer patients were consecutively recruited from the NSIA-LUTH Cancer Centre, Lagos, Nigeria, between September 2020 and February 2022. All participants completed the DT, BDI-II, and EORTC QLQ-C30/BR23 modules.

## Results

The mean DT score was 3.4 (SD ± 1.6), with 47.0% of participants meeting the NCCN threshold for clinically significant distress (DT ≥ 4). In contrast, only 15.9% had BDI-II scores ≥20, indicating moderate-to-severe depressive symptoms. The DT demonstrated diagnostic performance significantly worse than random classification, with an AUC of 0.414 (95% CI: 0.326–0.503), significantly below the threshold for acceptable diagnostic accuracy (p < 0.001). The Youden-optimal cutoff was DT ≥ 7.5, yielding a sensitivity of 2.0% and specificity of 98.5%. At the commonly used DT ≥ 4 threshold, sensitivity was 34.7% and specificity was 50.8%, indicating poor overall diagnostic utility.

## Conclusions

In this sub-Saharan African cancer cohort, the Distress Thermometer performed poorly in detecting clinically significant depression in this Nigerian MBC cohort when benchmarked against the BDI-II. This may reflect construct mismatch, somatic symptom confounding, cultural factors, or disease-specific characteristics of advanced cancer populations. The DT should not be used as a depression-specific screening tool in advanced cancer populations in LMICs, though its utility for identifying general distress remains unclear given the methodological limitations of this study. The poor concordance observed likely reflects construct mismatch, somatic symptom confounding in the reference standard, and the fundamental challenges of depression assessment in advanced cancer populations using self-report instruments. These findings underscore the critical need for appropriate reference standards (structured clinical interviews) and highlight methodological considerations in validating psychosocial screening tools across different constructs and cultural contexts.

## Introduction

Breast cancer represents the most frequently diagnosed malignancy worldwide and the leading cause of cancer-related mortality among women, accounting for approximately 11.7% of all new cancer cases and over 685,000 deaths annually [1]. The burden disproportionately affects women in low- and middle-income countries (LMICs), where late-stage presentation, limited treatment options, and poor survival outcomes compound the disease's impact [2]. Beyond its physical manifestations,

population, and ensures data availability for legitimate scientific purposes while preventing misuse or re-identification risks. Data Citation: Alabi, A.O. (Principal Investigator). Access to psychological support, psycho-oncology and psychotherapy among cancer patients in Nigeria [Dataset]. Lagos University Teaching Hospital, Lagos, Nigeria. Dataset available from: Lagos University Teaching Hospital Health Research Ethics Committee (LUTH-HREC), Idi-Araba, Lagos, Nigeria. Contact: cmd@lasuth.org; info@luth.gov.ng; +234 812 836 4824. IRB approval: ADM/DCST/HREC/APP/7058. Data collection period: September 2020 - February 2022.

**Funding:** The author(s) received no specific funding for this work.

**Competing interests:** The authors have declared that no competing interests exist.

breast cancer imposes profound psychological and emotional challenges, particularly among patients with metastatic disease who face an incurable prognosis and complex treatment trajectories.

Psychological distress, defined as a multifactorial emotional experience encompassing anxiety, depression, fear, and adjustment difficulties that can interfere with treatment adherence, quality of life, and survival, is increasingly recognized as a critical clinical outcome in oncology [3,4]. Among patients with metastatic breast cancer (MBC), prevalence rates of clinically significant distress range from 35% to 70%, substantially higher than those observed in early-stage disease [5]. Recognizing this burden, global cancer care guidelines now endorse routine distress screening, with many describing it as the "sixth vital sign" [6].

The integration of psychological screening into routine cancer care represents a paradigm shift toward comprehensive, patient-centered oncology. However, the effectiveness of such initiatives depends critically on the availability of validated, culturally appropriate screening instruments that can accurately identify patients requiring psychological intervention while remaining feasible for implementation in diverse healthcare settings [7].

One of the most widely adopted screening tools for distress is the Distress Thermometer (DT), developed by the National Comprehensive Cancer Network (NCCN) [8]. This instrument consists of a single-item visual analog scale ranging from 0 (no distress) to 10 (extreme distress), typically accompanied by a comprehensive problem checklist that identifies sources of physical, emotional, and practical concern. A DT score of ≥4 is commonly recommended to indicate clinically significant distress [9]. Due to its brevity, ease of administration, and integration potential into routine cancer care workflows, the DT has been widely adopted across oncology settings globally [10].

However, accumulating evidence suggests that the DT's diagnostic accuracy varies substantially across clinical populations, healthcare settings, and cultural contexts. Studies conducted in high-income countries (HICs), particularly those involving early-stage cancer patients, often report high sensitivity at the standard DT cutoff of ≥4 [9–11]. In contrast, investigations from LMICs and populations with advanced disease frequently report significantly lower sensitivity and markedly different optimal thresholds. For instance, a validation study conducted in Gaza found that a DT cutoff of ≥6 provided the optimal balance between sensitivity and specificity [12]. In Nigeria, a previous study in a mixed cancer population identified a cutoff of ≥3 that yielded 98% sensitivity but only 63% specificity [13].

In Nigeria specifically, two prior validation studies have evaluated the DT's performance, with encouraging findings. Lasebikan et al. [13] validated the DT in 130 cancer patients (mixed diagnoses and stages) using the Hospital Anxiety and Depression Scale (HADS) as reference standard, reporting an optimal cutoff of ≥4 with sensitivity of 89.3%, specificity of 73.7%, and an AUC of 0.87, indicating good diagnostic performance. Similarly, Obiajulu et al. [14] validated the DT in 90 ART-naïve HIV infected patients also using HADS, identifying an optimal cutoff of >5.0 with AUC range 0.754–0.709, again suggesting acceptable performance. These two Nigerian studies,

while showing some variation in optimal cutoffs, converge in suggesting that the DT has reasonable validity for detecting psychological distress in mixed Nigerian populations when using combined anxiety-depression measures as reference standards.

These discrepancies likely stem from complex sociocultural factors that influence how psychological distress is perceived, interpreted, and reported. In many African contexts, mental health stigma, collectivist coping mechanisms, spiritual attributions of suffering, and limited psychological literacy may significantly affect patients' willingness and ability to quantify emotional distress using Western-developed numeric scales [14,15]. Additionally, the linguistic and conceptual translation of "distress" across cultures presents inherent challenges, as equivalent terms may not exist or may carry different connotations in non-Western languages [16].

Despite the widespread adoption of the DT in global oncology practice, remarkably few validation studies have been conducted in sub-Saharan Africa, and none to date have specifically evaluated the instrument's performance in patients with metastatic breast cancer—a population particularly vulnerable to psychological morbidity due to treatment complexity, poor prognosis, and high symptom burden [17,18]. While studies from the United States have demonstrated that up to 60% of women with MBC report DT scores ≥4, [19] such findings may not generalize to African contexts, where psychosocial infrastructure is limited and cultural responses to suffering differ substantially from Western populations.

Given these prior validations suggesting adequate DT performance in mixed Nigerian cancer populations using combined anxiety-depression measures, the present study was designed to investigate whether these findings extend to the particularly vulnerable and distinct subgroup of patients with metastatic breast cancer. This population faces terminal prognosis, uniformly advanced disease, high physical symptom burden, and complex existential challenges that may influence psychological assessment differently than mixed-stage cohorts. Additionally, we examined DT performance using a depression-specific reference standard (BDI-II) rather than combined anxiety-depression measures (HADS), to evaluate the instrument's utility for identifying depression specifically, the most prevalent and treatable form of psychological morbidity in cancer care and the primary target for intervention in resource-limited settings. We hypothesized that the DT would demonstrate suboptimal diagnostic accuracy for identifying depression specifically (as distinct from general distress) in this population, due to: (1) construct mismatch between the broad, multidimensional distress captured by the DT and the specific depressive symptoms assessed by the BDI-II; and (2) potential somatic symptom confounding given the high physical symptom burden characteristic of metastatic cancer, which may inflate BDI-II scores independent of mood disturbance. This investigation provides important information about the DT's performance for a specific clinical application (depression screening) in advanced cancer populations, complementing prior Nigerian validation studies that examined general psychological distress using combined anxiety-depression measures. This study represents the first comprehensive evaluation of the diagnostic validity of the DT in identifying depressive symptoms among Nigerian women with metastatic breast cancer, using the Beck Depression Inventory-II (BDI-II) as the reference standard.

## Study objectives

The primary objective of this investigation was to evaluate the diagnostic accuracy of the Distress Thermometer in detecting clinically significant depressive symptoms among Nigerian women with metastatic breast cancer, using the Beck Depression Inventory-II (BDI-II) as the reference standard. Additionally, we sought to determine the optimal DT cutoff score for this population using Youden's Index and to examine the concordance between DT-defined distress and BDI-II-defined depression. We also aimed to identify clinical and demographic factors associated with discordant screening results between the two instruments, and ultimately to provide evidence-based recommendations for psychosocial screening approaches in LMIC oncology settings. These objectives were designed to address critical gaps in the psycho-oncology literature while informing clinical practice and health policy in resource-limited environments.

## Methods

### Study design and setting

This cross-sectional diagnostic validation study was conducted using de-identified data from the NSIA-LUTH Cancer Centre, a premier tertiary oncology facility located within Lagos University Teaching Hospital (LUTH), Nigeria. The center serves as a regional referral hub for cancer care in West Africa and offers comprehensive multidisciplinary services including medical oncology, radiation therapy, surgical oncology, palliative care, and limited psychosocial support services. The patient population predominantly comprises individuals from diverse socioeconomic backgrounds across Nigeria and neighboring West African countries. Data collection occurred over a 17-month period, from September 2020 to February 2022, encompassing both the initial COVID-19 pandemic period and subsequent recovery phases.

### Participants

Eligible participants comprised adult patients aged 18 years or older with histologically confirmed metastatic breast cancer (Stage IV disease according to AJCC 8th edition) and Eastern Cooperative Oncology Group (ECOG) performance status ranging from 0 to 3. Additional inclusion criteria required the ability to read and comprehend English or access to reliable translation services, intact cognitive functioning as assessed by clinical evaluation, and willingness to provide written informed consent. Patients were excluded from participation if they had active psychotic disorders or severe cognitive impairment that would preclude reliable self-report, were concurrently participating in psychological intervention studies, had missing data on primary outcome measures (DT or BDI-II scores), or were unable to complete questionnaires due to severe physical symptoms or medical instability. These criteria were designed to ensure a representative sample of the target population while maintaining data quality and participant safety.

### Sampling methodology

Consecutive sampling was employed to minimize selection bias and ensure representativeness of the target population. All eligible patients presenting for routine oncology care during the study period were approached for participation. Sample size was calculated based on an expected AUC of 0.75, with 80% power and $\alpha = 0.05$, requiring a minimum of 280 participants. To account for potential missing data and loss to follow-up, we aimed to recruit 320 participants.

### Measures

1. Distress thermometer (DT)

The Distress Thermometer represents a single-item, 11-point Likert-type visual analog scale ranging from 0 ("no distress") to 10 ("extreme distress"), designed to assess overall psychological burden experienced during the preceding week [8]. The DT was administered with the complete 34-item problem checklist organized into five domains: practical problems (transportation, insurance, work/school), family problems (dealing with children, dealing with partner), emotional problems (worry, fears, sadness, depression), spiritual/religious concerns, and physical problems (various symptoms). Participants were asked to indicate which problems they had experienced in the past week by checking applicable items across the five domains.

A threshold score of ≥4 has been widely endorsed by international guidelines, including those of the NCCN and the European Society for Medical Oncology (ESMO), to indicate clinically significant distress warranting further evaluation [9]. However, given documented variations in optimal cutoffs across populations and settings, we evaluated the full range of possible thresholds rather than assuming the appropriateness of this established cutoff.

The DT was selected for this investigation due to its brevity, ease of administration, widespread international adoption, and particular suitability for resource-limited environments where time constraints and limited mental health expertise may preclude the use of more complex screening instruments [20].

## 2. Beck depression inventory-II (BDI-II)

The Beck Depression Inventory-II represents a well-validated, 21-item self-report instrument designed to assess the severity of depressive symptoms over the preceding two-week period [21]. Individual items are scored on a 4-point scale (0–3), yielding total scores ranging from 0 to 63. The instrument demonstrates excellent psychometric properties, with reported Cronbach's alpha coefficients typically exceeding 0.90 in both clinical and non-clinical populations.

For this study, we employed a cutoff score of ≥20 to define clinically significant depressive symptoms, corresponding to moderate-to-severe depression severity. This threshold aligns with established guidelines for clinical practice and research applications, represents the level of symptom severity most likely to benefit from professional intervention, and has been validated in previous oncology studies [22,23].

We acknowledge that some validation studies in advanced cancer populations have identified lower optimal BDI-II cutoffs (e.g., ≥ 16) for screening purposes, which may increase sensitivity at the cost of specificity. However, we selected the ≥ 20 threshold to focus on moderate-to-severe depression most clearly warranting clinical intervention in resource-limited settings. Supplementary analyses using alternative BDI-II cutoffs (≥14, ≥ 19) are presented in Results to assess robustness of findings across different depression severity definitions.

We selected the BDI-II as our reference standard with full acknowledgment of the inherent construct mismatch in using a depression-specific instrument to validate a broad distress screening tool. This represents a significant methodological limitation, as discussed extensively in our limitations section. However, we pursued this design for specific pragmatic and clinical reasons, recognizing that our findings speak to the DT's performance for one particular clinical application (identifying patients with depression) rather than validating its full intended scope of measuring general distress. We made this methodological choice for several reasons. First, depression represents the most prevalent and clinically significant form of psychological morbidity in advanced cancer populations, and is the primary target for psychological intervention in most LMIC oncology settings where mental health resources are scarce. Second, while other DT validation studies have used broader measures such as the BSI-18, HADS total score, or DSM-IV interviews capturing multiple forms of distress, the BDI-II offers superior feasibility for implementation in resource-limited environments, requiring no specialized training for administration. Third, despite the DT's intended broad scope, clinical guidelines frequently recommend it for identifying patients with depression requiring referral for mental health services, making depression-specific validation clinically relevant. We explicitly acknowledge that our focus on depression as the outcome represents a narrower evaluation than the DT's full intended purpose, and that poor performance for depression detection does not necessarily indicate failure to capture other forms of distress. However, if the DT cannot adequately identify depression, the most common and treatable form of psychological morbidity in cancer, its clinical utility remains limited regardless of performance for other distress dimensions.

Nonetheless, we acknowledge that the ideal reference standard for depression validation in medically ill populations is a structured clinical interview (e.g., SCID, MINI) conducted by trained mental health professionals, which can distinguish depressive symptoms from somatic symptoms of cancer. The use of a self-report measure (BDI-II) as our reference standard, particularly one containing substantial somatic content, represents a significant limitation that may have contributed to the observed poor concordance. Our findings should be interpreted as reflecting the challenges of depression screening using brief self-report tools in advanced cancer populations, rather than definitively establishing the DT's cross-cultural invalidity.

## 3. EORTC QLQ-C30 and BR23 modules

To comprehensively characterize participants' health-related quality of life and contextualize psychological findings, all participants completed the European Organisation for Research and Treatment of Cancer Quality of Life Questionnaire

Core 30 (EORTC QLQ-C30) and its breast cancer-specific module (BR23) [24]. These extensively validated instruments evaluate multiple domains including physical functioning, role functioning, emotional well-being, cognitive functioning, social functioning, global health status, and disease-specific symptoms.

Scoring followed the official EORTC guidelines, employing linear transformation to generate standardized scores on a 0–100 scale. Higher scores on functional scales represent better functioning, while higher scores on symptom scales indicate greater symptom burden.

**Additional variables.** Comprehensive sociodemographic and clinical data were systematically collected to characterize the study population and identify potential confounding variables. Demographic characteristics included age, marital status, educational level, employment status, income level, and geographic origin. Disease-related variables encompassed histological subtype, hormone receptor status, HER2 status, sites of metastatic disease (including visceral, bone, and brain metastases), and the total number of metastatic sites. Treatment history documentation captured prior neoadjuvant therapy, surgical interventions, chemotherapy regimens, hormonal therapy, targeted therapy, and the number of previous treatment lines received. Performance status was assessed using the Eastern Cooperative Oncology Group (ECOG) scale at the time of enrollment. Psychosocial factors of interest included social support availability, religious and spiritual practices, and any history of prior mental health treatment. All clinical data were verified through comprehensive medical record review conducted by qualified oncology physicians to ensure accuracy and completeness.

## Statistical analysis

**Descriptive analysis.** Continuous variables are presented as means with standard deviations or medians with interquartile ranges, depending on distributional characteristics. Categorical variables are reported as frequencies and percentages. Normality of continuous variables was assessed using the Shapiro-Wilk test and visual inspection of histograms and Q-Q plots.

**Diagnostic accuracy analysis.** The primary analysis employed receiver operating characteristic (ROC) curve methodology to evaluate the DT's diagnostic performance relative to the BDI-II reference standard. The area under the ROC curve (AUC) serves as a summary measure of diagnostic accuracy, with values approaching 1.0 indicating excellent discrimination and values around 0.5 suggesting performance no better than chance [25]. We calculated 95% confidence intervals for the AUC using DeLong's method and tested the null hypothesis that AUC = 0.5 using the Mann-Whitney U statistic.

**Optimal cutoff determination.** The optimal DT cutoff score was determined using Youden's Index (J = sensitivity + specificity − 1), which identifies the threshold that maximizes the sum of sensitivity and specificity [26]. This approach is particularly appropriate when the costs of false positives and false negatives are considered approximately equal, as is often the case in screening applications.

For each potential cutoff value, we calculated Sensitivity (true positive rate), Specificity (true negative rate), Positive predictive value (PPV), Negative predictive value (NPV), Positive likelihood ratio (PLR), Negative likelihood ratio (NLR), and Diagnostic odds ratio (DOR).

**Additional analyses.** Several supplementary analyses were conducted to enhance the robustness of our findings and explore potential sources of variation in diagnostic performance. Correlation analysis between DT and BDI-II scores was performed using Pearson's correlation coefficient to quantify the strength of association between these measures. Subgroup analyses were stratified by age groups, performance status categories, and metastatic disease sites to examine whether diagnostic accuracy varied across clinically relevant patient subpopulations. We evaluated alternative BDI-II cutoff thresholds (≥14, ≥19, and ≥29) to assess the consistency of our findings across different definitions of clinically significant depression. Additionally, we analyzed individual DT problem checklist items and their associations with BDI-II scores to better understand which specific domains of distress might be most predictive of depressive symptoms in this population.

## Statistical software and significance

All analyses were conducted using R statistical software version 4.4.1 (R Core Team (2024). R: A Language and Environment for Statistical Computing. R Foundation for Statistical Computing, Vienna, Austria). Specific packages included pROC for ROC analysis, psych for reliability analysis, and corrplot for correlation matrices. Statistical significance was set at $\alpha = 0.05$ for all tests. Given the exploratory nature of subgroup analyses, we applied Bonferroni correction where appropriate to control for multiple testing.

## Missing data management

Missing data were minimal (< 5% for any variable) and appeared to occur at random based on Little's MCAR test. Complete case analysis was performed for the primary diagnostic accuracy analysis to preserve the integrity of sensitivity and specificity estimates. Sensitivity analyses using multiple imputation were conducted to assess the robustness of findings.

## Ethical considerations

The study protocol received approval from the Lagos University Teaching Hospital Health Research Ethics Committee (LUTH-HREC, IRB ADM/DCST/HREC/APP/7058). All participants provided written informed consent after receiving comprehensive information about study procedures, potential risks and benefits, and their right to withdraw at any time without affecting their medical care. Special considerations were made for participants with limited literacy, including the availability of witnessed verbal consent procedures when appropriate.

This study represents a secondary analysis of de-identified patient data originally collected for a broader investigation of quality of life and psychosocial outcomes in Nigerian women with metastatic breast cancer. All identifying information was removed prior to data sharing, and researchers had access only to anonymized data. Data were first accessed for this secondary analysis on 27/10/2023.

Data confidentiality and security were maintained in accordance with international guidelines, including the Declaration of Helsinki and Good Clinical Practice standards. All data were de-identified and stored on password-protected servers with restricted access limited to authorized study personnel.

# Results

## Participant characteristics and study flow

A total of 313 patients were initially screened for study participation, of whom 309 (98.7%) met all eligibility criteria and completed the required assessments ([Fig 1]). Four participants (1.3%) were excluded due to incomplete data on primary outcome measures. The high retention rate reflects the robustness of our recruitment strategy and the acceptability of study procedures to participants.

The study population comprised predominantly female participants (98.7%, n = 305), with a median age of 53.0 years (interquartile range: 44.0–63.0 years). The age distribution was slightly right-skewed, reflecting the inclusion of some younger patients with aggressive disease. Educational attainment varied considerably, with 34.6% having completed secondary education and 22.3% having obtained post-secondary qualifications. Approximately 45.0% of participants reported being unemployed or unable to work due to their illness, highlighting the substantial socioeconomic impact of metastatic disease.

## Primary outcome measures

**Distress thermometer findings.** The mean DT score across the entire sample was 3.4 (SD ± 1.6, range: 0–8). The distribution was approximately normal with a slight positive skew. Using the internationally recommended threshold of DT ≥ 4, 145 participants (47.0%) were classified as experiencing clinically significant distress. Notably, no participant

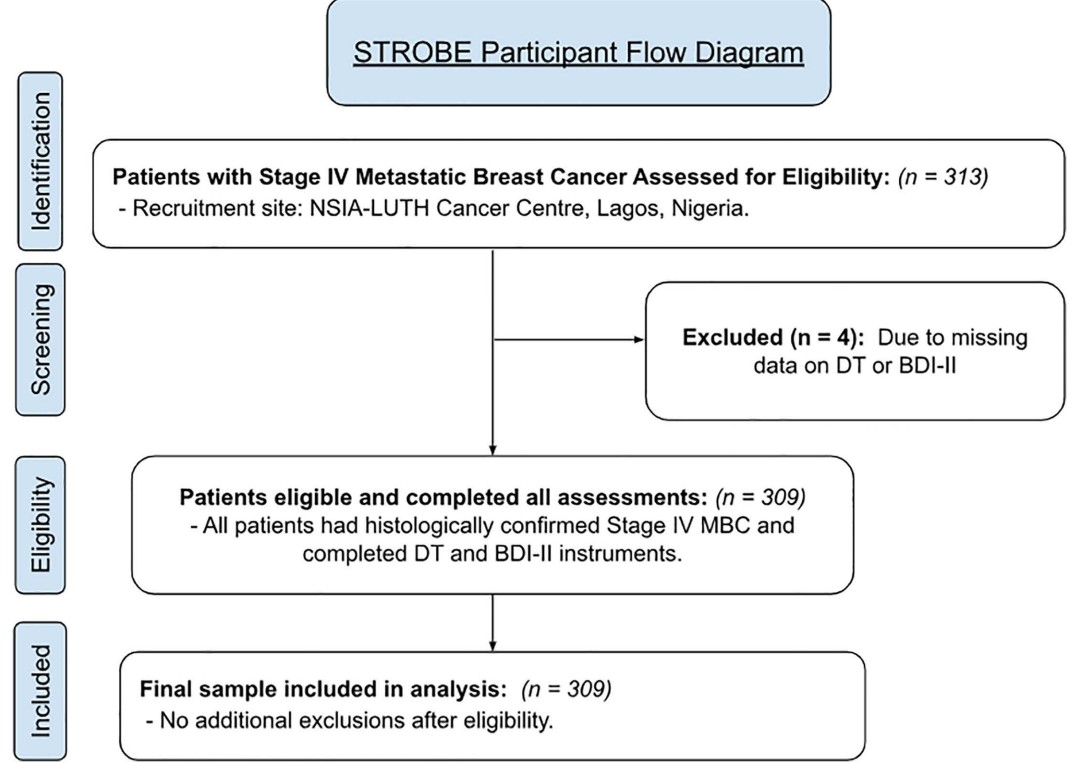

**Fig 1. STROBE-compliant participant flow diagram.** Study participant recruitment and enrollment flow. A total of 313 patients with metastatic breast cancer were screened for eligibility at NSIA-LUTH Cancer Centre, Lagos, Nigeria (September 2020–February 2022). Four patients were excluded due to incomplete primary outcome data (DT or BDI-II scores), resulting in 309 participants included in the final analysis. The high retention rate (98.7%) demonstrates the feasibility and acceptability of study procedures in this population.

reported the maximum distress score of 10, and only 2 participants (0.6%) reported a score of 9. Only 12 participants (3.9%) reported scores of 8 or higher, suggesting possible cultural factors affecting the use of extreme scale anchors. Our rate of clinically significant distress (47.0% with DT ≥ 4) is somewhat lower than the 60% or higher rates reported in Western studies of women with metastatic breast cancer [19], though this difference may reflect cultural response patterns, different study periods, or genuine differences in distress prevalence rather than measurement validity issues alone.

**Beck depression inventory-II findings.** BDI-II scores averaged 12.3 (SD ± 7.2, range: 0–39), with the distribution demonstrating positive skewness consistent with a predominantly non-depressed sample. Applying our predetermined cutoff of BDI-II ≥ 20 for clinically significant depressive symptoms, only 49 participants (15.9%) met this criterion (**Table 1**). An additional 95 participants (30.7%) scored in the mild depression range (BDI-II 14–19), while 130 participants (42.1%) demonstrated minimal or no depressive symptoms (BDI-II 0–13).

**Discordance between measures.** Substantial discordance was observed between DT-defined distress and BDI-II-defined depression, consistent with these instruments measuring different psychological phenomena. Of the 145 participants with DT ≥ 4, only 33 (22.8%) also met criteria for clinically significant depression (BDI-II ≥ 20). Conversely, 17 of the 49 participants (34.7%) with BDI-II ≥ 20 scored below the DT threshold for distress. This substantial lack of concordance (κ = 0.12, indicating poor agreement) suggests that these instruments are measuring distinct, albeit related, psychological constructs. It is also plausible that distress in this population is driven more by physical and practical

**Table 1. Baseline demographic, clinical, psychosocial, and quality of life characteristics of participants (N = 309).**

| Demographics | |
|---|---|
| Age, years (Median) | 53.0 (IQR: 44.0–63.0) |
| Sex, n (%) | Female: 305 (98.7%)<br>Male: 4 (1.3%) |
| **Clinical Characteristics** | |
| Distress Thermometer Score, mean ± SD | 3.4 ± 1.6 |
| Clinically Significant Distress (DT ≥ 4), n (%) | 145 (47.0%) |
| Beck Depression Inventory Score, mean ± SD | 12.3 ± 7.2 |
| Clinically Significant Depression (BDI ≥ 20), n (%) | 49 (15.9%) |
| Visceral Metastasis Present, n (%) | 213 (69.0%) |
| Spine Metastasis Present, n (%) | 151 (48.9%) |
| Brain Metastasis Present, n (%) | 40 (12.9%) |
| Received Neoadjuvant Chemotherapy, n (%) | 62 (20.1%) |
| Received Any Chemotherapy, n (%) | 176 (57.0%) |
| Had Surgery, n (%) | 149 (48.2%) |
| Received Hormone Therapy, n (%) | 13 (4.2%) |
| Received Immunotherapy, n (%) | 0 (0%) |
| **Psychosocial Problem Scores** | |
| Practical Problems Count, mean ± SD | 2.7 ± 1.6 |
| Emotional Problems Count, mean ± SD | 2.3 ± 1.9 |
| Physical Problems Count, mean ± SD | 5.0 ± 3.6 |
| **Quality of Life Scores (Standardized)** | |
| Physical Symptom Score, mean ± SD | 31.3 ± 17.4 |
| Emotional Symptom Score, mean ± SD | 19.0 ± 13.3 |
| Global Quality of Life Score, mean ± SD | 62.7 ± 11.1 |

DT = Distress Thermometer; BDI = Beck Depression Inventory-II. Clinically significant distress (DT ≥ 4) and depression (BDI ≥ 20) were defined per NCCN guidelines and moderate symptom severity thresholds, respectively. Quality of life scores were standardized using EORTC QLQ-C30/BR23 scoring manuals.

problems (captured by the DT) than by the specific psychological construct of depression (assessed by the BDI-II), though both instruments are likely confounded by somatic symptoms in this advanced cancer population.

## Clinical and disease characteristics

Regarding metastatic disease patterns, 213 participants (69.0%) presented with visceral metastases, 151 (48.9%) had spinal involvement, and 40 (12.9%) had brain metastases. Multiple metastatic sites were common, with 156 participants (50.5%) having ≥3 distinct sites of disease. Brain metastasis was present in 40 (12.9%), pleural metastasis in 57 (18.4%), while liver metastases occurred in 42 (13.6%) participants.

Treatment history revealed that 176 patients (57.0%) had received chemotherapy, 149 patients (48.2%) had undergone surgery, and 62 patients (20.1%) had received neoadjuvant therapy. Hormonal therapy utilization was surprisingly low at 4.2% (n = 13), likely reflecting the predominance of triple-negative breast cancer in this population, consistent with epidemiological patterns observed in sub-Saharan Africa. No participants had received immunotherapy, reflecting financial barriers and limited access to novel therapeutic agents in this LMIC setting.

EORTC QLQ-C30 assessments revealed substantial symptom burden, with mean Physical Symptom Scores of 31.3 (SD ± 17.4) and Emotional Symptom Scores of 19.0 (SD ± 13.3). The Global Quality of Life Score averaged 62.7

(SD ± 11.1), indicating moderate impairment relative to population norms. Problem checklist analysis revealed that participants reported an average of 5.0 physical problems (SD ± 3.6), 2.7 practical problems (SD ± 1.6), and 2.3 emotional problems (SD ± 1.9), with physical symptoms representing the predominant source of distress (Table 1).

### Diagnostic performance analysis

**ROC curve analysis.** The ROC curve analysis revealed extremely poor diagnostic performance of the DT in identifying participants with clinically significant depressive symptoms (Fig 2). The AUC was 0.414 (95% CI: 0.326–0.503), significantly below the threshold for acceptable diagnostic accuracy (p < 0.001 for comparison with AUC = 0.5). This poor concordance between the DT and BDI-II in identifying participants with clinically significant depressive symptoms likely reflects that these instruments measure different psychological constructs in this population. As discussed in our limitations section, this pattern is consistent with the DT capturing primarily physical and practical distress while the BDI-II, containing substantial somatic content, is confounded by cancer-related symptoms in this advanced-stage population.

**Optimal cutoff analysis.** Using Youden's Index to optimize the balance between sensitivity and specificity, the optimal DT cutoff was identified as ≥7.5. However, this threshold yielded extremely poor sensitivity (2.0%) while achieving high specificity (98.5%). The positive predictive value was 20.0%, and the negative predictive value was 84.2%. The positive likelihood ratio was 1.33 and the negative likelihood ratio was 1.00, both indicating minimal diagnostic utility (Fig 3).

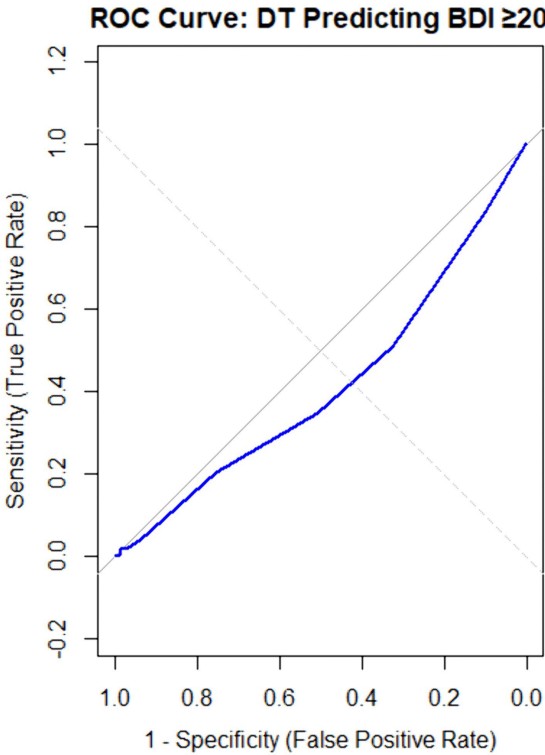

**Fig 2. Receiver operating characteristic (ROC) curve showing the performance of the Distress Thermometer (DT) in predicting clinically significant depressive symptoms (BDI ≥ 20).** ROC curve analysis demonstrating the diagnostic performance of the Distress Thermometer for detecting clinically significant depressive symptoms (BDI-II ≥ 20) in Nigerian women with metastatic breast cancer (n = 309). The AUC was 0.414 (95% CI: 0.326–0.503), indicating performance significantly worse than chance (diagonal reference line, AUC = 0.5). The curve's position below the reference line confirms the DT's inadequate discriminatory ability for depression screening in this population.

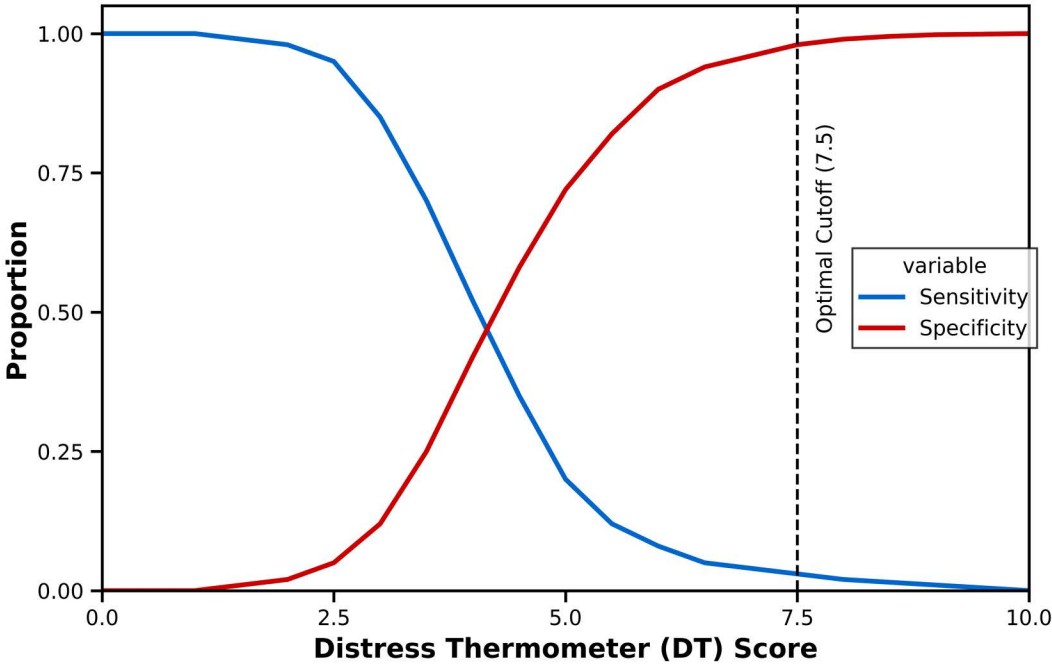

**Fig 3. Sensitivity and specificity by DT cutoff scores.** Sensitivity and specificity curves for the Distress Thermometer across all possible cutoff values (0.5–9.5) for detecting clinically significant depressive symptoms (BDI-II ≥ 20). The red line represents sensitivity and the blue line represents specificity. The vertical dashed line indicates the Youden-optimal cutoff (DT = 7.5), which achieved 98.5% specificity but only 2.0% sensitivity. The curves demonstrate the classic inverse relationship between sensitivity and specificity, with no threshold providing clinically acceptable performance (≥80% sensitivity and ≥70% specificity simultaneously).

**Performance across multiple thresholds.** Examination of diagnostic performance across all possible DT cutoffs revealed no clinically acceptable threshold (**Table 2**). The conventional DT ≥ 4 cutoff demonstrated sensitivity of 34.7% and specificity of 50.8%, representing suboptimal performance by any clinical standard. Even lowering the threshold to DT ≥ 1.5 achieved sensitivity of only 83.7% with specificity of 10.0%, resulting in an unacceptably high false-positive rate.

The diagnostic odds ratio (DOR) for the optimal cutoff (DT ≥ 7.5) was 1.34 (95% CI: 0.18–10.2), indicating minimal discriminatory ability. Across all evaluated thresholds, no combination achieved both clinically acceptable sensitivity (≥80%) and specificity (≥70%) simultaneously.

## Subgroup and supplementary analyses

**Correlation and concordance analysis.** The Pearson correlation coefficient between DT and BDI-II total scores was $r = 0.23$ (95% CI: 0.12–0.34, $p < 0.001$), indicating a weak positive association that explains only 5.3% of the variance between measures. This finding suggests that while the instruments demonstrate some convergent validity, they are largely measuring distinct psychological constructs. The modest correlation coefficient is consistent with the substantial discordance observed between DT-defined distress and BDI-II-defined depression in our primary analysis, reinforcing the conclusion that these tools are not interchangeable for clinical assessment purposes.

**Subgroup analyses by patient characteristics.** Comprehensive subgroup analyses revealed remarkably consistent poor diagnostic performance across all major patient demographic and clinical characteristics, suggesting that the DT's limitations in this population are not confined to specific patient subgroups (**Fig 4**). Among younger patients (≤50 years), the AUC was 0.38 compared to 0.44 in older patients (>50 years), with no

**Table 2. Diagnostic performance of distress thermometer (DT) cutoffs for predicting clinically significant depression (BDI ≥ 20).**

| DT Cutoff | Sensitivity | Specificity | Youden Index | PPV* | NPV* | PLR | NLR |
|---|---|---|---|---|---|---|---|
| 0.5 | 1 | 0.004 | 0.004 | 0.159 | 1 | 1.00 | 0.00 |
| 1.5 | 0.837 | 0.1 | −0.063 | 0.149 | 0.765 | 0.93 | 1.63 |
| 2.5 | 0.51 | 0.327 | −0.163 | 0.125 | 0.78 | 0.76 | 1.50 |
| 3.5 | 0.347 | 0.508 | −0.145 | 0.117 | 0.805 | 0.71 | 1.29 |
| 4.5 | 0.204 | 0.754 | −0.042 | 0.135 | 0.834 | 0.83 | 1.06 |
| 5.5 | 0.041 | 0.938 | −0.021 | 0.111 | 0.838 | 0.66 | 1.02 |
| 6.5 | 0.02 | 0.969 | −0.010 | 0.111 | 0.84 | 0.65 | 1.01 |
| **7.5** | **0.02** | **0.985** | **0.005** | **0.2** | **0.842** | **1.33** | **1.00** |
| 8.5 | 0 | 0.988 | −0.012 | 0 | 0.84 | 0.00 | 1.01 |
| 9.5 | 0 | 0.992 | −0.008 | 0 | 0.84 | 0.00 | 1.01 |

Youden's Index identified DT ≥ 7.5 as the optimal cutoff, yielding high specificity (98.5%) but extremely poor sensitivity (2.0%), precluding clinical utility. The NCCN-recommended threshold of DT ≥ 4 demonstrated inadequate diagnostic performance in this population. PPV = Positive Predictive Value; NPV = Negative Predictive Value; PLR = Positive Likelihood Ratio; NLR = Negative Likelihood Ratio. **Clinical Interpretation Note:** The optimal cutoff (DT ≥ 7.5) has PLR = 1.33 and NLR = 1.00, both indicating minimal diagnostic value. Clinically useful likelihood ratios are typically >10 for PLR (to rule in disease) and <0.1 for NLR (to rule out disease). Negative Youden Index values indicate that the test performs worse than random chance at those cutoffs, with sensitivity + specificity summing to less than 1.0. This reflects the inverse relationship observed between DT and BDI-II in this population.

statistically significant difference between groups (p = 0.45). This finding indicates that age-related factors such as coping mechanisms, disease perception, or communication styles do not substantially influence the DT's diagnostic utility. Similarly, performance status did not meaningfully affect diagnostic accuracy, with patients having better functional status (ECOG 0–1) demonstrating an AUC of 0.41 compared to 0.40 in those with poorer performance status (ECOG 2–3, p = 0.89). This suggests that physical functioning and symptom burden, while potentially influencing psychological well-being, do not alter the fundamental measurement properties of the DT in this population.

Analyses stratified by metastatic disease patterns revealed that patients with visceral metastases had an AUC of 0.39 compared to 0.46 in those with non-visceral disease (p = 0.34), again demonstrating no clinically meaningful differences. The consistently poor performance across disease severity indicators suggests that the DT's limitations are not attributable to specific disease characteristics but rather reflect more fundamental issues with the instrument's validity in this cultural and clinical context.

**Alternative cutoff analysis.** Using alternative BDI-II cutoffs yielded similar results. AUC = 0.43 (95% CI: 0.36–0.50) for BDI-II ≥ 14 and AUC = 0.51 (95% CI: 0.40–0.62) for BDI-II ≥ 29, confirming the robustness of our primary findings.

### Relationship between distress, depression, and quality of life

To better understand the discordance between DT and BDI-II measurements, we examined their relationships with quality of life domains assessed by the EORTC QLQ-C30. The DT demonstrated moderate positive correlation with physical symptom burden (r = 0.34, p < 0.001) and weak correlation with emotional functioning (r = 0.19, p = 0.001). In contrast, the BDI-II showed stronger correlation with emotional functioning (r = 0.42, p < 0.001) and weaker correlation with physical symptoms (r = 0.28, p < 0.001). Both instruments showed similar modest correlations with global quality of life (DT: r = −0.26; BDI-II: r = −0.31; both p < 0.001). Analysis of DT problem checklist domains revealed that participants with elevated BDI-II scores (≥20) endorsed significantly more emotional problems (mean 3.8 vs 2.0, p < 0.001) but not significantly more physical problems (mean 5.4 vs 4.9, p = 0.34) compared to those with BDI-II < 20. These patterns suggest that the DT and BDI-II capture distinct aspects of patient experience, with the DT more heavily influenced by physical symptomatology while the BDI-II more specifically reflects emotional distress.

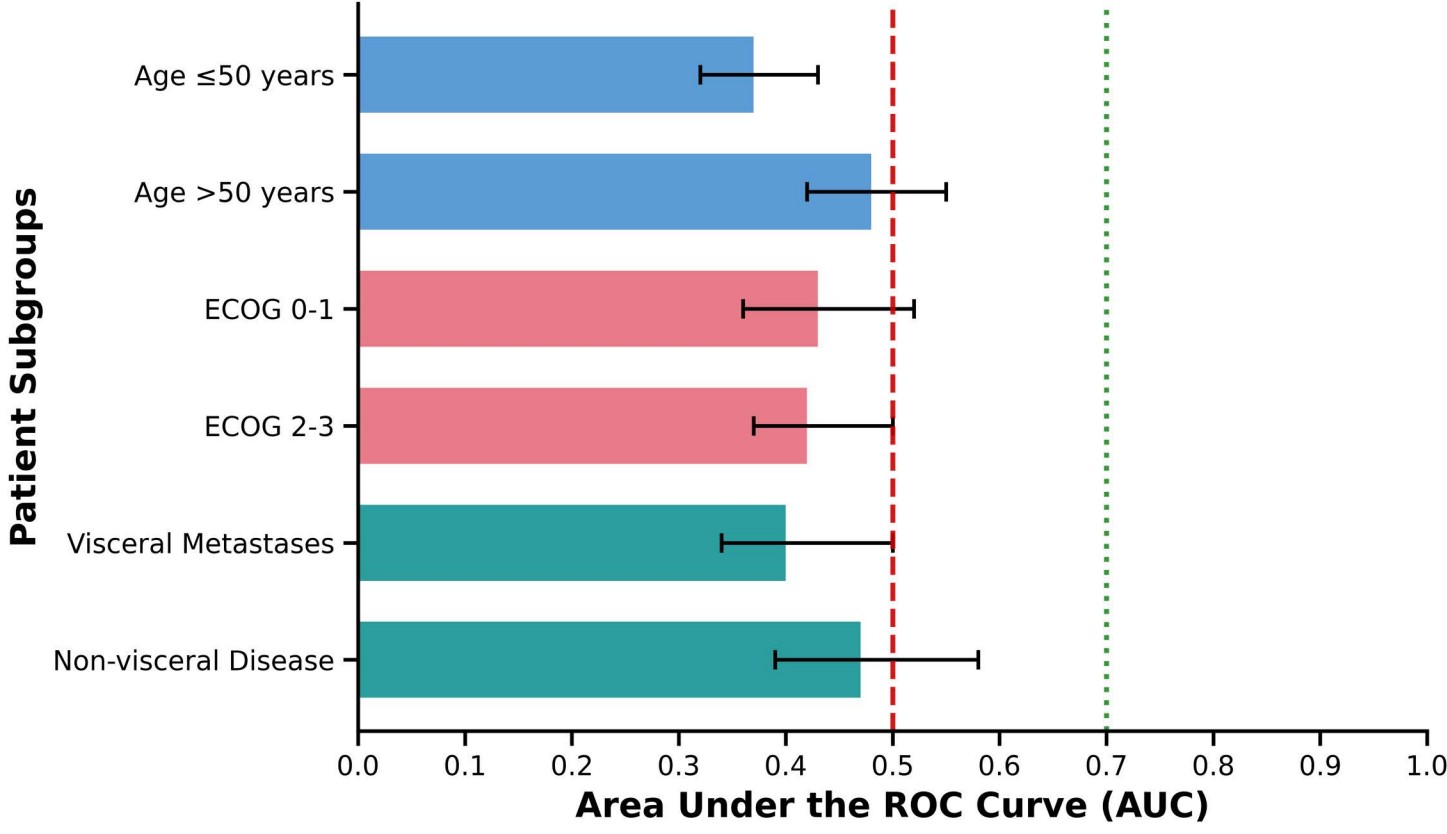

**Fig 4. Forest plot showing the area under the receiver operating characteristic curve (AUC) for the Distress Thermometer's ability to detect clinically significant depressive symptoms (BDI-II ≥ 20) across different patient subgroups.** Error bars represent 95% confidence intervals. The horizontal dashed line at AUC = 0.5 indicates performance no better than chance, while the dotted line at AUC = 0.7 represents the threshold for acceptable diagnostic accuracy. All subgroups demonstrated poor diagnostic performance with AUC values well below the acceptable threshold, indicating consistent failure of the DT across patient characteristics. P-values compare AUC between subgroups within each category using DeLong's test for correlated ROC curves. Reference lines: Dashed line (AUC = 0.5) = chance performance; Dotted line (AUC = 0.7) = acceptable diagnostic accuracy threshold. Statistical note: No subgroup achieved clinically acceptable diagnostic performance (AUC ≥ 0.7). The consistently poor performance across all demographic and clinical characteristics suggests fundamental limitations of the DT in this population rather than subgroup-specific issues.

## Discussion

### Principal findings and clinical implications

This study represents the first comprehensive validation of the Distress Thermometer in sub-Saharan African patients with metastatic breast cancer and reveals concerning findings regarding its utility for depression-specific screening. Our results demonstrate poor concordance between the DT and BDI-II for identifying depressive symptoms, likely reflecting methodological limitations rather than systematic failure of the DT as a screening instrument. With an AUC of 0.414, the instrument performed significantly worse than random classification, challenging fundamental assumptions about its universal applicability in cancer care.

The magnitude of diagnostic failure observed in our study is particularly striking when contextualized against international validation studies. While previous investigations in high-income countries typically report AUC values ranging from 0.70 to 0.85 for the DT [27,28], our findings suggest a complete breakdown of the instrument's psychometric properties in this LMIC setting. The optimal cutoff of DT ≥ 7.5, despite achieving high specificity, demonstrated very

poor sensitivity (2.0%), indicating inadequate performance for depression-specific screening in this methodological context.

Perhaps most concerning is the substantial discordance between DT-defined distress (47.0% of participants) and BDI-II-defined depression (15.9% of participants). This finding suggests that nearly half of patients reporting significant distress on the DT do not meet criteria for clinically significant depression, while conversely, many patients with depression are not identified as distressed by the DT.

## Somatic symptom overlap and reference standard limitations

Beyond the construct mismatch between broad distress and depression-specific measurement discussed above, a critical methodological factor that may substantially explain our findings is the confounding effect of somatic symptoms in advanced cancer populations. The most parsimonious explanation for our findings, including the extraordinary AUC below 0.5, is the fundamental methodological artifact created by using a somatically-confounded, depression-specific self-report instrument to validate a broad distress screening tool in an advanced cancer population. This methodological issue likely contributes more substantially to the observed poor concordance than cultural factors or cross-cultural invalidity of the DT itself. The BDI-II, while an excellent depression screening instrument in general populations, contains seven items (33% of total) that assess somatic/neurovegetative symptoms: fatigue/loss of energy (Item 15), changes in sleeping pattern (Item 16), irritability (Item 17), changes in appetite (Item 18), concentration difficulty (Item 19), tiredness or fatigue (Item 20), and loss of interest in sex (Item 21). These symptoms overlap extensively with the direct physiological effects of advanced cancer and its treatment.

In our MBC population experiencing uniformly advanced disease with high physical symptom burden (mean EORTC Physical Symptom Score 31.3 ± 17.4), participants likely endorsed many BDI-II somatic items due to cancer-related symptoms rather than depression *per se*. This phenomenon, namely, somatic symptom inflation of depression scores in medically ill populations, is well-documented in the psycho-oncology literature but particularly pronounced in advanced cancer. A patient with severe cancer-related fatigue, chemotherapy-induced sleep disruption, pain-related concentration difficulties, and disease-related appetite loss may score in the moderate depression range on the BDI-II (total score 15–20 or higher) without experiencing the core mood and cognitive symptoms of clinical depression.

Critically, this somatic confounding may have created a systematic bias in our reference standard that explains the inverse relationship we observed (AUC < 0.5). If many participants had elevated BDI-II scores driven primarily by somatic symptoms rather than true depression, while the DT (being a single-item global rating) was more accurately capturing psychological distress specifically, this would produce the pattern we observed, DT and BDI-II measuring different phenomena and showing poor concordance or even inverse relationships at certain cutoffs.

This interpretation is supported by our finding that the DT correlated more strongly with physical symptom burden than with emotional functioning, suggesting participants were primarily rating physical/somatic distress when completing the thermometer. Meanwhile, the BDI-II, intended to measure depression, was paradoxically also capturing substantial somatic symptomatology. The result is two instruments that, in this advanced cancer population with overwhelming physical symptoms, were both contaminated by somatic burden but in different ways, producing poor concordance that reflects methodological artifact rather than true diagnostic failure of either instrument.

This sobering realization highlights a fundamental limitation of our study design: we compared one screening tool (DT) against another screening tool (BDI-II), rather than against a gold-standard diagnostic interview conducted by a mental health professional who could determine whether somatic symptoms represented depression, cancer effects, or both. A diagnostic interview allows clinical judgment to weigh the relative contribution of mood *versus* disease burden to reported symptoms, a nuance impossible with self-report questionnaires alone. Our AUC of 0.414 therefore reflects, at least in part, the discordance between two imperfect instruments both affected by somatic confounding, rather than necessarily indicating poor validity of the DT *per se*.

This methodological explanation does not entirely exculpate the DT, which still failed to identify depression adequately in our sample, but it substantially reframes the interpretation. Rather than representing a catastrophic failure of cross-cultural validity, our findings may primarily reflect the methodological challenges of depression assessment in advanced cancer using self-report screening tools, a challenge that would likely affect any brief instrument in this population. Future validation studies in advanced cancer populations must employ diagnostic clinical interviews as gold standards to disentangle these confounded relationships.

## Construct mismatch and measurement specificity

A critical consideration in interpreting our findings concerns the fundamental difference between what the DT and BDI-II measure. The Distress Thermometer was designed as a broad screening tool to capture multiple dimensions of distress including anxiety, depression, existential concerns, practical problems, spiritual distress, and physical symptom burden. In contrast, the BDI-II specifically measures depressive symptoms. This construct mismatch raises the question of whether our findings reflect cultural invalidity of the DT or simply the expected consequence of using a depression-specific reference standard to validate a broad distress measure.

We acknowledge this methodological tension explicitly. The poor AUC we observed (0.414) may partly reflect the DT's capture of non-depressive forms of distress that are not measured by the BDI-II. Our supplementary analysis of EORTC QLQ-C30 relationships provides evidence for this interpretation: the DT correlated more strongly with physical symptom burden ($r = 0.34$) than emotional functioning ($r = 0.19$), while the BDI-II showed the opposite pattern ($r = 0.42$ for emotional functioning vs $r = 0.28$ for physical symptoms). This suggests the DT may be more responsive to somatic distress while the BDI-II captures psychological distress more specifically.

However, we contend that this construct mismatch, rather than invalidating our conclusions, actually underscores a critical clinical limitation of the DT in LMIC oncology settings. Current clinical guidelines, including those from NCCN and ESMO, recommend the DT not merely as a general distress screening tool but specifically for identifying patients with anxiety and depression requiring mental health referral. In resource-limited settings where comprehensive psychological assessment is often unavailable, the DT is frequently used as the primary or sole mechanism for identifying patients needing psychological intervention, and depression represents the most prevalent and treatable form of psychological morbidity in these contexts. If the DT cannot adequately discriminate patients with clinically significant depression from those without, its utility as a practical clinical tool is fundamentally compromised regardless of its ability to capture "distress" in some broader, less clinically actionable sense.

Moreover, the pattern of discordance we observed suggests the DT may be identifying distress driven primarily by physical symptom burden and practical problems rather than psychological morbidity per se. This is evidenced by the high proportion of participants with elevated DT scores but low BDI-II scores, and by the stronger correlation between DT and physical symptoms compared to emotional functioning. While physical and practical concerns certainly merit clinical attention, they require different interventions than depression (symptom management, social services) and should not be conflated with psychiatric morbidity. A screening tool that cannot distinguish between somatic distress and depression has limited utility for guiding referral decisions in settings where mental health resources must be carefully allocated.

Therefore, while we acknowledge our study evaluated the DT against a depression-specific rather than broad distress reference standard, we maintain that our findings reveal clinically significant inadequacies in the instrument's performance for its primary intended purpose in cancer care settings.

## Construct non-equivalence and cross-cultural measurement challenges

Beyond the immediate diagnostic accuracy findings, our results raise fundamental questions about construct validity and measurement equivalence in cross-cultural psycho-oncology research. The substantial discordance observed between DT-defined "distress" and BDI-II-defined "depression" in our Nigerian cohort (47.0% vs. 15.9%) may reflect not merely

measurement error or tool inadequacy, but rather fundamental differences in how these psychological phenomena are conceptualized, experienced, and expressed across cultural contexts.

In Western psychological frameworks, "distress" and "depression" represent related but distinguishable constructs, both falling within an individualistic model of emotional suffering as internal psychological states amenable to numeric quantification. However, in many African cultural contexts, emotional suffering may be understood through fundamentally different frameworks, as spiritual experiences, social disruptions, or somatic manifestations, that do not map neatly onto Western psychological categories. What Western instruments label as "distress" or "depression" may be experienced and articulated by Nigerian patients as ancestral displeasure, family disharmony, or bodily imbalance.

This construct non-equivalence has profound implications for the validity of cross-cultural research in psycho-oncology. Our findings suggest that simply translating and administering Western-developed instruments, even those with strong psychometric properties in their original contexts, may yield measurements of uncertain meaning when the underlying psychological constructs themselves may not exist or may be structured differently across cultures. The poor correlation between DT and BDI-II scores in our sample ($r = 0.23$) supports this interpretation, indicating that these instruments may be tapping into qualitatively different dimensions of experience in this population.

## Cultural and contextual factors influencing DT performance

Several interconnected factors likely contribute to the DT's poor performance in this Nigerian cohort. While the cultural interpretations we offer below are plausible and consistent with existing literature on mental health in African contexts, we acknowledge that without formal cognitive debriefing or qualitative assessment of how participants understood and responded to the DT, these explanations remain speculative. Direct evidence from patients about their subjective experience of completing the instrument would be necessary to definitively determine the mechanisms underlying the DT's poor performance. First, cultural conceptualizations of psychological distress may differ fundamentally from Western frameworks underlying the instrument's development. In many African societies, emotional suffering is often understood through spiritual, communal, or somatic lenses rather than individualistic psychological models [29,30]. The concept of rating one's internal emotional state on a numeric scale may be culturally unfamiliar or inappropriate, leading to inconsistent or invalid responses.

Second, pervasive mental health stigma in Nigerian society may lead to systematic underreporting of psychological symptoms [31]. Participants may fear discrimination, family rejection, or perceived weakness if they acknowledge significant emotional distress, particularly in a medical setting where physical symptoms are prioritized. This phenomenon could explain why many participants with clinically significant depression (as measured by the more detailed BDI-II) report low distress scores on the single-item DT.

Third, linguistic and semantic challenges may compromise the validity of distress measurement. While our study was conducted in English, many participants likely think and feel in their native languages (Yoruba, Igbo, Hausa, among others), where concepts of "distress" may not have direct equivalents or may carry different connotations. The translation of internal emotional experiences into English-language numeric ratings may introduce systematic measurement error.

Fourth, the collectivist cultural orientation prevalent in Nigerian society may influence how individuals conceptualize and report personal distress. In collectivist cultures, emotional experiences are often understood in relational and community contexts rather than as individual psychological states, potentially rendering single-item self-report measures less valid.

## Comparison with international literature

Our findings align with a growing body of evidence questioning the universal applicability of Western-developed psychosocial screening instruments. A systematic review of DT validation studies across diverse populations revealed substantial heterogeneity in optimal cutoff scores and diagnostic performance metrics [32]. Studies conducted in non-Western

populations consistently reported lower sensitivity and altered optimal thresholds compared to investigations in high-income Western countries.

Specifically, while DT validation studies using broad distress measures (BSI-18, HADS total score) in high-income countries have reported AUC values of 0.70–0.85 [33], depression-specific validations show more variable performance with AUC values frequently in the 0.60–0.70 range. our AUC of 0.414 for depression detection remains substantially lower than even these more modest benchmarks, suggesting fundamental challenges beyond the typical variance observed across validation studies. Studies in other LMIC settings have shown intermediate performance with AUC values of 0.60–0.70 for various psychological outcomes [8,34]. This pattern suggests a gradient of diagnostic utility that correlates with cultural and socioeconomic distance from the instrument's development context.

The discordance between our findings and those from high-income countries cannot be attributed solely to differences in disease stage or treatment setting. Previous studies of advanced cancer patients in Western settings, while showing somewhat reduced DT performance compared to early-stage populations, have not demonstrated the complete diagnostic failure observed in our cohort. This suggests that cultural and contextual factors, rather than disease-related variables, represent the primary drivers of poor performance.

## Comparison with prior Nigerian validation studies

Our findings present a striking contrast to prior Nigerian DT validations and require careful interpretation. Lasebikan et al. [13] and Obiajulu et al. [14] both reported good-to-excellent diagnostic performance (AUCs of 0.87 and 0.82 respectively) in a mixed-stage Nigerian cancer population and a non-cancer cohort, respectively, while our study found diagnostic performance significantly worse than chance (AUC 0.414). This dramatic discordance demands systematic exploration of potential explanatory factors.

Several factors may explain the discordance between these prior studies and our findings. Regarding implementation methodology, both Lasebikan et al. [13] and Obiajulu et al. [14] administered instruments in controlled research settings with standardized protocols, though neither study provided detailed descriptions of specific administration procedures, interviewer training protocols, or quality control measures beyond their use of HADS as reference standard. Our study similarly employed standardized administration protocols with trained research personnel, suggesting that differences in basic implementation methodology are unlikely to fully explain the divergent findings. First, both previous studies used HADS, which measures combined anxiety and depression, whereas we used the depression-specific BDI-II. The DT may perform better at detecting general psychological distress (anxiety plus depression) than depression specifically, consistent with its design as a broad screening tool. Second, our population consisted exclusively of patients with meta-static breast cancer, a uniformly advanced-stage cohort facing terminal prognosis, whereas Lasebikan et al. [13] included mixed cancer stages and Obiajulu et al. [14] studied non-cancer patients. Disease severity and proximity to death may influence how distress is experienced and reported. Third, our sample was 98.7% female, while the prior studies included more balanced gender distributions; cultural factors around emotional expression may differ between men and women in Nigerian society. Fourth, the different reference standards may be measuring genuinely different psychological constructs, our focus on moderate-to-severe depression (BDI-II $\geq$ 20) represents a more specific and severe outcome than general psychological distress measured by HADS.

Rather than contradicting these prior Nigerian studies, our findings complement them by providing the first depression-specific validation in an advanced cancer population. Taken together, the three Nigerian studies suggest that while the DT may have some utility for detecting general psychological distress in mixed Nigerian populations, its performance deteriorates substantially when (a) the outcome of interest is depression specifically rather than combined anxiety/depression, and (b) the population consists of patients with advanced, terminal illness. These nuanced interpretation has important clinical implications. In Nigerian oncology settings, the DT may retain utility for initial broad screening in mixed cancer populations, identifying patients with any form of significant distress who warrant further evaluation. However, it

cannot be relied upon as a depression-specific screening tool, particularly in advanced cancer populations where somatic symptoms confound assessment. If depression identification is the clinical goal, as it often is in resource-limited settings where treatment is targeted at the most prevalent and treatable condition, then depression-specific instruments or clinical interviews are necessary following positive DT screens.

## Methodological strengths and study rigor

This investigation possesses several methodological strengths that enhance confidence in our findings. First, to the best of our knowledge, our study represents the largest validation of the DT in a sub-Saharan African cancer population and the only investigation specifically focused on metastatic breast cancer in this region. The homogeneous disease population eliminates confounding from cancer stage or treatment heterogeneity that has complicated previous multi-cancer validation studies.

Second, we employed rigorous diagnostic accuracy methodology, including comprehensive ROC curve analysis, multiple cutoff evaluation, and optimal threshold determination using established statistical methods. Our use of Youden's Index, while conventional, was supplemented by examination of clinical utility indices and likelihood ratios to provide a comprehensive assessment of diagnostic performance.

Third, the BDI-II represents an appropriate and well-validated reference standard for depression screening in cancer populations. Unlike some previous DT validation studies that used clinical interviews or other distress measures as reference standards, our choice of a depression-specific instrument allows for clearer interpretation of discordant results.

Fourth, our comprehensive assessment of potential confounding variables, including detailed clinical and sociodemographic data, allows for robust interpretation of findings and identification of factors that might influence screening performance.

## Study limitations and interpretive considerations

Several limitations warrant acknowledgment in interpreting our findings. First, and most critically, we did not conduct formal cognitive debriefing or qualitative assessment of how participants interpreted and responded to the DT scale. This represents a critical gap in our investigation, as cognitive interviews could have revealed whether patients struggled with the numeric rating format, misunderstood the concept of "distress," interpreted the scale anchors differently than intended, or experienced cultural barriers to reporting psychological symptoms. Without this qualitative component, we cannot definitively determine why the DT failed in this population, whether due to semantic/linguistic issues, cultural inappropriateness of the rating format, stigma-related response bias, or fundamental construct non-equivalence. While our cultural interpretations are plausible and consistent with existing literature on mental health in African contexts, there is need for direct evidence from participants about their subjective experience of completing the instrument. Future validation studies in LMIC settings should employ mixed-methods designs that combine quantitative diagnostic accuracy assessment with qualitative cognitive interviewing to understand the mechanisms underlying tool performance or failure. Such approaches would not only identify whether an instrument works but also why it succeeds or fails, providing crucial insights for cultural adaptation efforts and tool development.

Second, an important limitation concerns our English language inclusion criterion. While we allowed access to translation services, requiring English comprehension may have systematically excluded patients with lower educational attainment or from rural areas who might express and report psychological distress differently. This language requirement may have resulted in a sample with greater familiarity with Western conceptual frameworks and numeric rating scales, potentially underestimating the cultural barriers to valid DT administration in the broader Nigerian population. Our findings may therefore represent a "best-case scenario" for DT performance, with even poorer diagnostic accuracy likely in populations with more limited English proficiency or formal education. Future studies should include participants assessed in their native languages to provide a more comprehensive picture of the DT's cross-cultural validity.

Third is the limitation of using one screening instrument (BDI-II) as the reference standard for validating another screening instrument (DT), rather than employing a diagnostic gold-standard clinical interview conducted by a mental health professional. While the BDI-II represents a well-validated depression screening instrument, it remains a self-report measure rather than a structured clinical interview. However, the BDI-II's strong psychometric properties and widespread validation in cancer populations support its appropriateness as a reference standard for this investigation. Additionally, we acknowledge that our use of a depression-specific reference standard to validate a broad distress measure represents a methodological limitation, as discussed extensively in our construct mismatch section. The poor performance we observed may partly reflect this measurement incongruence rather than solely cultural invalidity.

Fourth, the high potential for somatic symptom overlap in the BDI-II to have confounded our results represents a critical interpretive limitation. Seven of the 21 BDI-II items (33%) assess somatic/neurovegetative symptoms (fatigue, sleep disturbance, appetite changes, concentration difficulty, loss of interest in sex, irritability, tiredness) that overlap extensively with the direct physiological effects of advanced cancer and its treatment. In our metastatic breast cancer population with uniformly high physical symptom burden (mean EORTC Physical Symptom Score $31.3 \pm 17.4$), BDI-II scores were likely inflated by cancer-related symptoms independent of mood disturbance. This somatic confounding may have created a reference standard that systematically misclassifies patients, some with high BDI-II scores driven by somatic symptoms without true depression, others with low BDI-II scores despite depression because psychological symptoms are over-shadowed by overwhelming physical symptoms. If the DT was more accurately capturing psychological distress while the BDI-II was confounded by somatic symptoms, this would produce the inverse relationship we observed (AUC < 0.5) through methodological artifact rather than true DT failure. The absence of a diagnostic interview prevents us from determining the true depression prevalence in our sample and from validating whether the BDI-II ≥ 20 threshold identified genuine depression or somatic symptom burden. This fundamentally limits our ability to conclude that the DT "failed", as it may have been responding to a different (and potentially more valid) construct than our confounded reference standard. Future studies must employ clinical interviews as gold standards in advanced cancer populations to disentangle these relationships.

Fifth, our cross-sectional design precludes assessment of how screening performance might vary across different points in the cancer trajectory or in response to treatment changes. Longitudinal investigations would provide valuable insights into the temporal stability of DT performance and its relationship to disease and treatment dynamics. Moreso, while our sample was drawn from Nigeria's largest cancer center and likely represents the broader population of patients with MBC in the region, generalizability to other African countries or LMIC settings cannot be assumed. Cultural, linguistic, and healthcare system differences across LMICs may influence screening instrument performance in unpredictable ways. Moreover, our findings are specific to MBC and may not be generalized to earlier-stage disease or other cancer types without validation in those populations.

Sixth, our focus on depression as the primary outcome may not capture the full spectrum of psychological morbidity relevant to cancer care. The DT was designed as a broad screening tool for multiple forms of distress, and its performance in identifying anxiety, adjustment disorders, or existential distress was not evaluated in this study.

## Clinical practice implications and actionable recommendations for LMIC oncology settings

The clinical implications of our findings are profound and immediate. Oncology providers in LMIC settings should exercise extreme caution when using the DT as a stand-alone screening tool for psychological morbidity, particularly depression. Our results suggest that reliance on the DT with standard cutoffs will result in missing the majority of patients with clinically significant depressive symptoms while potentially over-identifying patients without such symptoms.

Based on our findings, we propose a structured, multi-stage screening approach for psychological morbidity in LMIC cancer care settings. The first stage involves implementing universal brief screening using locally validated instruments administered to all patients at key clinical timepoints, including diagnosis, treatment initiation, treatment completion, and

disease progression. Until culturally appropriate alternatives are developed, clinicians may use the DT cautiously as a conversation starter rather than a diagnostic tool, recognizing its profound limitations for identifying depression. The DT may retain utility for initiating discussions about sources of distress, particularly when used alongside the problem check-list to identify practical, physical, and social concerns that require attention regardless of depression status.

The second stage requires that all patients screening positive on brief tools, as well as those identified through clinical observation as potentially distressed, undergo semi-structured clinical interviews conducted by trained oncology staff. These interviews should focus on culturally relevant expressions of distress rather than relying exclusively on Western psychological terminology such as "depression" or "anxiety." Specifically, interviewers should explore somatic complaints and physical symptom preoccupation, changes in social and family functioning, spiritual concerns and religious coping mechanisms, practical problems including financial hardship and transportation barriers, and disturbances in sleep and appetite patterns. This culturally grounded approach acknowledges that emotional suffering may manifest through physical, social, and spiritual channels in African contexts rather than solely through psychological symptomatology.

In the third stage, patients identified through clinical interview as requiring intervention should receive comprehensive psychological assessment using culturally adapted, multi-dimensional instruments. When feasible and culturally appropriate, collateral information from family members can provide valuable contextual understanding, given the collectivist orientation prevalent in many African societies where individual distress is understood within relational and communal frameworks. Referral to mental health specialists should occur when such resources are available, though task-shifting models utilizing trained nurses or social workers may be necessary in resource-limited settings where psychiatric expertise remains scarce.

Successful implementation of such multi-stage screening requires substantial investment in training and capacity building. Healthcare systems must train oncology nurses, social workers, and physicians to recognize culturally specific expressions of psychological distress, understand how somatization, spiritual attributions, and collectivist coping may manifest in African cancer patients, and conduct brief clinical interviews that explore distress through culturally relevant frameworks. Training should emphasize cultural humility and the importance of avoiding the imposition of Western psychological categories onto patient experiences. Staff must also develop competence in identifying patients requiring referral for specialized mental health care while recognizing the limitations of available resources.

Screening programs must be designed for feasibility within existing cancer care workflows to ensure sustainable implementation. This necessitates conducting brief screening at routine clinic visits rather than requiring separate appointments that burden already overwhelmed patients and healthcare systems. Screening documentation must be seamlessly integrated into existing medical records, with clear protocols established for positive screen follow-up and referral. Regular team meetings should be scheduled to review screening results, discuss complex cases, and ensure continuity of care across the oncology team.

Critical to the success of any screening program is the consideration of resource allocation and ethical implementation. Given the limited mental health infrastructure in many LMIC settings, screening programs must be coupled with realistic referral pathways and adequate intervention capacity. Implementing screening without available treatment resources raises profound ethical concerns and may paradoxically increase patient distress by identifying problems without offering solutions. Healthcare systems should therefore conduct comprehensive resource mapping to identify available mental health services, develop task-shifting models to expand intervention capacity beyond specialized psychiatry, train oncology staff in basic psychological support techniques, establish formal partnerships with psychiatric services and community mental health resources, and consider group interventions and peer support programs as cost-effective approaches to expanding treatment access. Only through such comprehensive planning can screening programs fulfill their promise of improving patient outcomes rather than merely documenting unmet needs.

Healthcare systems should also invest in training oncology staff to recognize cultural expressions of psychological distress that may not be captured by Western-developed screening tools. This includes understanding how patients from

different cultural backgrounds may express emotional suffering through somatic complaints, spiritual concerns, or social disruption rather than direct psychological terminology.

The substantial discordance between DT and BDI-II results in our study highlights the non-interchangeable nature of different psychological constructs. Clinicians should recognize that "distress" and "depression" represent distinct albeit related phenomena, and that screening approaches should be tailored to the specific psychological outcomes of greatest clinical concern.

## Health policy and global guidelines implications

Our findings have significant implications for international cancer control guidelines and health policy in LMICs. Currently, major organizations including the NCCN, ESMO, and WHO recommend routine distress screening using the DT with relatively uniform cutoff recommendations [35,36]. Our results challenge the evidence base underlying these recommendations and suggest that blanket adoption of Western-developed tools without local validation may be ineffective or potentially harmful.

We recommend that international cancer organizations fundamentally revise their psychosocial screening guidelines to explicitly acknowledge that instruments developed and validated in high-income Western populations may not perform adequately in LMIC or non-Western settings, thereby removing prescriptive recommendations for uniform tool adoption across diverse cultural contexts.

Guidelines should mandate local validation studies before implementing any screening program, providing clear frameworks that specify minimum methodological standards, recommended sample sizes, appropriate reference standards for different cultural contexts, and statistical approaches for determining optimal cutoffs. Rather than promoting reliance on single screening instruments, guidelines should emphasize multi-stage, multi-method approaches that combine brief screening tools with clinical interviews and culturally relevant assessment methods, particularly in LMIC settings where Western psychological frameworks may not apply. International organizations should actively support the development of regional and national psychosocial screening guidelines that reflect local cultural contexts, healthcare system capacities, and patient populations, while providing technical assistance and funding for guideline development processes in LMICs. Training recommendations must include specific guidance on cultural competence and recognition of culturally specific expressions of psychological distress.

Corresponding shifts in funding priorities are essential to support these guideline revisions. Research funding should prioritize indigenous tool development and validation studies in LMICs, community-based participatory research that engages local stakeholders in screening tool design, and implementation science examining the integration of culturally appropriate screening into existing cancer care workflows. International agencies should establish collaborative networks for psycho-oncology research across diverse cultural settings and develop standardized protocols for cross-cultural validation studies that can be adapted to local contexts while maintaining methodological rigor.

LMICs should consider developing regional or national guidelines for psychosocial screening that reflect local cultural contexts, healthcare system capacities, and patient populations. Such guidelines should emphasize flexible, multi-modal approaches to psychological assessment rather than reliance on any single screening instrument.

## Research implications and future directions

This investigation opens several critical avenues for future research in psycho-oncology and global health. First, urgent efforts are needed to develop and validate culturally appropriate psychological screening tools for African cancer populations using community-based participatory research approaches. Such development should engage patients, families, survivors, and community members throughout the process, while including traditional healers and religious leaders

who often serve as frontline mental health resources in these contexts. Oncology healthcare providers familiar with local contexts must be involved in identifying culturally relevant indicators of psychological distress through qualitative methods including focus groups and in-depth interviews. Development processes should explore preferred assessment formats ranging from numeric scales to pictorial representations and narrative approaches, testing comprehension and acceptability across diverse educational and linguistic groups to ensure broad applicability.

Second, future validation research must employ integrated mixed-methods designs that address the critical limitation identified in our study regarding the absence of cognitive debriefing. Such studies should incorporate cognitive interviewing to understand how patients interpret screening questions and whether their interpretation matches intended meaning, focus groups exploring cultural conceptualizations of psychological distress and barriers to reporting symptoms, ethnographic observation of patient-provider communication to identify conversational patterns and cultural norms, think-aloud protocols where patients verbalize their thought processes while completing instruments, and member checking to validate researcher interpretations with community stakeholders. These qualitative components are essential for understanding not merely whether screening tools work, but why they succeed or fail in specific cultural contexts.

Third, comparative effectiveness research should systematically evaluate different screening strategies, including single-instrument approaches, multi-stage screening models, and clinician-based assessment without formal instruments, and hybrid models combining cultural and Western screening methods. Outcomes should encompass detection rates for various forms of psychological morbidity, false-positive and false-negative rates, referral patterns and treatment initiation, cost-effectiveness and resource requirements, patient satisfaction and acceptability, and ultimate impact on quality of life and clinical outcomes. Longitudinal studies investigating the temporal stability of screening instrument performance across the cancer trajectory would provide crucial insights into optimal timing and frequency of psychological assessment, how screening performance varies with disease progression or treatment changes, and the natural history of psychological distress in LMIC cancer populations.

Fourth, the ultimate value of improved screening depends on the availability of effective interventions. Research must test whether enhanced psychological screening translates into improved clinical outcomes, quality of life, and patient satisfaction, while evaluating culturally adapted psychological interventions for African cancer populations. Studies should examine task-shifting models where non-specialists deliver mental health support and assess the feasibility and effectiveness of group-based and peer support interventions as scalable approaches.

Finally, implementation science research should examine broader healthcare system requirements including training needs and curricula for oncology staff, referral pathway development and optimization, integration strategies with existing cancer care workflows, barriers and facilitators to screening program implementation, and considerations for sustainability and scalability in resource-constrained settings.

## Theoretical implications for psycho-oncology

Our findings contribute to broader theoretical discussions about the universality of psychological constructs and measurement instruments in psycho-oncology. The failure of a widely validated Western instrument in our African population raises fundamental questions about whether psychological distress represents a universal human experience that can be measured consistently across cultures, or whether distress is culturally constructed in ways that render cross-cultural measurement problematic.

The substantial discordance between DT and BDI-II results in our study, despite both instruments ostensibly measuring psychological morbidity, highlights the multidimensional and culturally contingent nature of emotional suffering. Rather than representing a unitary construct, "distress" appears to encompass multiple distinct phenomena including depression, anxiety, existential concerns, practical problems, social disruption, and spiritual crisis that may require different assessment approaches and may be differentially salient across cultural contexts.

Our findings align with anthropological critiques of psychiatric universalism that question whether diagnostic categories developed in Western contexts represent genuine natural kinds versus culture-bound constructs that emerge from specific historical, philosophical, and social frameworks. The poor correlation between DT and BDI-II in our Nigerian sample (r = 0.23) raises the possibility that these instruments, rather than measuring the same underlying construct with different degrees of precision, may actually be tapping into qualitatively different dimensions of experience that are structured differently across cultural contexts.

This has profound implications for the validity of cross-cultural research in psycho-oncology. Simply translating and administering instruments cross-culturally, even when accompanied by careful linguistic validation, may yield measurements of uncertain meaning if the underlying psychological constructs themselves are not equivalent across cultures. The concept of "measurement invariance", the assumption that an instrument measures the same construct in the same way across groups, may not hold when comparing Western and non-Western populations, necessitating fundamental rethinking of how we assess psychological phenomena across cultures.

These findings support calls for more nuanced, culturally grounded approaches to psycho-oncology research and practice that recognize the diversity of human responses to cancer across different cultural contexts [37,38]. Rather than seeking to validate Western instruments across cultures with the goal of achieving measurement equivalence, psycho-oncology may need to embrace a more pluralistic approach that develops culture-specific frameworks and assessment methods while maintaining rigorous scientific standards.

## Conclusions

This study demonstrates that the Distress Thermometer, despite its widespread international endorsement and ease of use, is not suitable for depression-specific screening in advanced cancer populations when validated against depression-focused self-report measures. With an AUC of 0.414, the instrument performed significantly worse than random classification in this unique methodological and clinical context, suggesting systematic failure rather than merely suboptimal performance. However, the interpretation of these findings requires careful consideration of multiple explanatory factors that extend beyond simple cross-cultural invalidity.

Our results diverge markedly from two prior Nigerian validations showing good DT performance in mixed-stage cancer and non-cancer populations using combined anxiety-depression measures (HADS) as reference standards. This discordance likely reflects methodological differences, specifically, our use of a depression-specific rather than broad distress reference standard, our focus on uniformly advanced-stage disease with high somatic symptom burden, and the fundamental limitation of validating one screening tool against another rather than against diagnostic clinical interviews. The substantial somatic symptom overlap in both instruments (DT capturing physical distress, BDI-II including somatic depression items) may have created confounding that explains much of the poor concordance observed, particularly in our metastatic population with overwhelming physical symptoms.

This study demonstrates that the Distress Thermometer should not be used as a depression-specific screening tool in advanced cancer populations in LMICs, though its utility for identifying general distress remains unclear and requires validation using appropriate reference standards. The poor concordance with the BDI-II observed in this study most likely reflects the methodological challenges of validating a broad distress measure against a depression-specific, somatically-confounded self-report instrument in an advanced cancer population, rather than definitively establishing the DT's cross-cultural invalidity.

Our study's design, using one self-report instrument to validate another, beclouds the possibility of determining with certainty whether the poor performance reflects construct mismatch, somatic symptom confounding, cultural factors, or some combination thereof. However, the most parsimonious interpretation, given the extraordinary AUC < 0.5 and the substantial somatic overlap in our reference standard, is that methodological artifacts may have substantially contributed to the observed findings. These results underscore the critical importance of using structured clinical

interviews as gold standards for depression validation in medically ill populations and highlight the need for methodologically rigorous validation studies that appropriately match screening tools with reference standards measuring the same construct.

Our results demonstrate that effective psychosocial screening in oncology requires more than simply translating and implementing validated instruments from high-income countries. Instead, meaningful progress demands recognition of cultural, methodological, and contextual factors that influence how psychological distress is experienced, expressed, and reported across diverse populations.

This investigation represents a crucial first step toward developing evidence-based, culturally appropriate approaches to psychological assessment in sub-Saharan African cancer populations. However, substantial additional research and health system development will be required to translate these insights into improved clinical care. The broader significance of this work lies not in definitively proving DT failure, but in demonstrating the critical importance of rigorous, context-specific validation before implementing screening tools in new populations or clinical contexts. Our outlier findings relative to prior Nigerian studies underscore how methodological choices (reference standard selection, population characteristics, outcome definition) fundamentally shape validation results and clinical conclusions.

The implications extend beyond Nigeria to the broader global cancer community, highlighting the urgent need for local validation of imported health technologies and the development of contextually appropriate alternatives. Only through such efforts can we ensure that psychosocial oncology advances reach the patients who need them most, regardless of geographic location or cultural background.

Our findings represent an important contribution to understanding the complexities of cross-cultural psychometric validation, particularly in vulnerable populations with advanced illness. However, we emphasize that our findings are specific to depression screening in metastatic breast cancer using a particular reference standard, and cannot be generalized to all forms of distress, all cancer stages, all LMIC settings, or all screening contexts. The field requires additional studies employing diagnostic interviews, cognitive debriefing, and careful attention to somatic confounding to build a more comprehensive evidence base for psychosocial assessment in diverse global oncology populations.

The complexity of psychological assessment in cancer care, particularly across cultural contexts and disease stages, demands nuanced interpretation and cumulative evidence rather than definitive proclamations based on individual investigations. We call upon international cancer organizations, funding agencies, researchers, and healthcare providers to prioritize the development and validation of culturally sensitive psychosocial screening approaches for LMIC cancer populations. The failure to address these disparities perpetuates global inequities in comprehensive cancer care and undermines efforts to achieve universal health coverage and cancer control.

## What is already known about the Topic?

i. The Distress Thermometer (DT) is widely used as a rapid screening tool for psychological morbidity in cancer care, particularly in high-income countries.

ii. Most validation studies have focused on early-stage cancer or mixed cancer populations in high-resource settings, with limited evidence from low- and middle-income countries (LMICs).

iii. There is substantial variation in optimal DT cutoff scores across populations, settings, and cultures, underscoring the need for local validation.

iv. Limited data exist on the diagnostic validity of the DT among patients with metastatic breast cancer in sub-Saharan Africa.

**What this paper adds**

i. This is the first study to evaluate the diagnostic accuracy of the DT in detecting clinically significant depressive symptoms among Nigerian women with metastatic breast cancer, using the Beck Depression Inventory-II (BDI-II) as the reference standard.

ii. The DT demonstrated extremely poor diagnostic performance, with an area under the ROC curve (AUC) of 0.414, indicating that it performed worse than random classification in this population.

iii. The substantial discordance between DT-defined distress (47.0%) and BDI-II-defined depression (15.9%) highlights the non-interchangeable nature of these constructs in this population.

iv. The Youden-optimal cutoff of DT ≥ 7.5 yielded high specificity (98.5%) but extremely poor sensitivity (2.0%), highlighting a failure to detect most true cases of depression.

v. Even at lower thresholds, the DT failed to achieve clinically acceptable diagnostic performance, with no threshold providing both adequate sensitivity (≥80%) and specificity (≥70%).

**Implications for practice, theory, or policy**

i. Clinicians in LMIC oncology settings should exercise caution when using the DT as a stand-alone depression screening tool in patients with advanced cancer, and consider implementing multi-stage screening approaches.

ii. Uniform DT cutoffs recommended by global guidelines (e.g., DT ≥ 4) may not be appropriate in culturally diverse or resource-limited contexts.

iii. Health systems in LMICs should invest in the development and validation of culturally sensitive screening instruments and hybrid models that combine brief tools with targeted clinical interviews.

iv. International cancer organizations should revise global guidelines to emphasize the need for local validation before implementing standardized psychosocial screening tools.

v. This study supports calls for contextual adaptation of psycho-oncologic tools and the incorporation of culturally grounded approaches to psychological assessment in global cancer control strategies.

## Author contributions

**Conceptualization:** Abdulrazzaq Lawal, Anthonia Sowunmi, Adewumi Alabi, Oluwaseun Adebayo Bamodu.

**Data curation:** Tejen Chavda, Abdulrazzaq Lawal, Anthonia Sowunmi, Bolanle Adegboyega, Adewumi Alabi, Chen-Chih Chung, Sylvia Shirima, Donaldson F. Conserve, Oluwaseun Adebayo Bamodu.

**Formal analysis:** Tejen Chavda, Bolanle Adegboyega, Chen-Chih Chung, Sylvia Shirima, Donaldson F. Conserve, Oluwaseun Adebayo Bamodu.

**Funding acquisition:** Adewumi Alabi.

**Investigation:** Abdulrazzaq Lawal, Anthonia Sowunmi, Bolanle Adegboyega, Adewumi Alabi.

**Methodology:** Adewumi Alabi, Oluwaseun Adebayo Bamodu.

**Project administration:** Anthonia Sowunmi, Adewumi Alabi, Donaldson F. Conserve, Oluwaseun Adebayo Bamodu.

**Resources:** Adewumi Alabi, Donaldson F. Conserve.

**Software:** Tejen Chavda.

**Supervision:** Oluwaseun Adebayo Bamodu.

**Validation:** Oluwaseun Adebayo Bamodu.

**Visualization:** Tejen Chavda, Oluwaseun Adebayo Bamodu.

**Writing – original draft:** Tejen Chavda, Oluwaseun Adebayo Bamodu.

**Writing – review & editing:** Adewumi Alabi, Oluwaseun Adebayo Bamodu.

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
