## [Decision Letter · Decision Letter 0]

15 Sep 2025

Dear Dr. Bamodu,

Thank you for submitting your manuscript to PLOS ONE. After careful consideration, we feel that it has merit but does not fully meet PLOS ONE’s publication criteria as it currently stands. Therefore, we invite you to submit a revised version of the manuscript that addresses the points raised during the review process. Please see comments below.

We look forward to receiving your revised manuscript.

Kind regards,

Alejandro Botero Carvajal, Ph.D

Academic Editor

PLOS ONE

Journal Requirements:

2. In the online submission form, you indicated that [De-identified participant data may be made available upon reasonable request and approval by the institutional ethics committees, subject to data sharing agreements that ensure participant privacy and appropriate use of data.].

Reviewers' comments:

Reviewer's Responses to Questions

**Comments to the Author**

1. Is the manuscript technically sound, and do the data support the conclusions?

Reviewer #1: Yes

Reviewer #2: Partly

Reviewer #3: Partly

2. Has the statistical analysis been performed appropriately and rigorously?

Reviewer #1: Yes

Reviewer #2: Yes

Reviewer #3: Yes

3. Have the authors made all data underlying the findings in their manuscript fully available?

Reviewer #1: No

Reviewer #2: No

Reviewer #3: Yes

4. Is the manuscript presented in an intelligible fashion and written in standard English?

Reviewer #1: Yes

Reviewer #2: Yes

Reviewer #3: Yes

Reviewer #1: Your paper addresses an important gap, and the negative findings are valuable for practice and policy. The use of BDI-II as a reference and the statistical methods are rigorous, and the writing is generally clear with detailed methods and discussion.

That said, a few areas could be strengthened:

The cultural interpretation is central to your argument, but you did not include cognitive debriefing or qualitative assessment of how patients understood the DT. This is more than a minor limitation and deserves stronger emphasis in the discussion.

Similarly, the paper should acknowledge more explicitly that “distress” and “depression” may not be equivalent constructs across cultures. This would sharpen your interpretation and situate your findings in a broader theoretical context.

Consider expanding your discussion of solutions. You already note the limitations of the DT, but readers would benefit from more detail on possible alternatives (e.g., hybrid screening approaches, locally adapted tools).

Providing clearer, actionable guidance for oncology practitioners in LMICs would enhance the clinical relevance of your work.

Minor issues:

Line 195–196: “usig” → should be “using.”

Line 367: missing closing parenthesis before “suggests.”

Line 378: “financial barrier” → should be “financial barriers.”

Several references need correction:

Ref 3 appears incomplete or inconsistent (Lancet Oncology citation).

Ref 4 (NCCN guidelines) is misformatted as a journal article. Please revise according to journal style.

Reviewer #2: The authors present interesting work evaluating the distress thermometer as compared to the BDII depression score. The topic is important as guidelines promote DT use and claim "global" validation (https://onlinelibrary.wiley.com/doi/full/10.1002/pon.3430) but validation (at least initially) was limited. The study is well-powered and asks very important questions about cultural adaptation of screening tools in LMICs. However, a few issues limit the strength of the conclusions as they are stated. While limitations are well discussed in the final section, the wording and conclusions prior are not consistent with this section. Additionally, the authors do not make the underlying data available which is typically a requirement of PLOS ONE.

Methods:

Clarify whether DT was administered with the full problem list or only the thermometer item. This affects interpretability as the DT with inventory can shape perception.

It is unclear where the Psychosocial Problem Scores reported in Table 1 originate from? Which instrument?

Figures/Tables: Figures are generally clear. Consider reporting likelihood ratios alongside sensitivity/specificity for clinical interpretability.

Results: The authors report the collection of EORTC-QOL information, but the relationship between DT and symptom burden, or BDII and symptom burden is not evaluated. It is unclear why this is omitted as it may provide clarity wrt the discordance of the results.

Discussion:

Additional inclusion criteria required the ability to read and comprehend English or access to reliable translation services - the authors mention the lack of data on the interpretation of the BDII and DT but the authors should also raise this limitation and the potential impact on the final cohort or results.

As the authors mention in the final paragraph, the distress thermometer (DT) scale is designed to measure a combination of distress factors (existential, practical, anxiety, depressive, physical) whereas the BDII score is focused on depressive symptoms alone. However, the authors claim that the BDII is a good reference despite the lack of use as a reference standard in other validation studies which have used the BSI-18, Brief Symptom Inventory-18, DSM-IV, and HADS, Hospital Anxiety and Depression Scale (https://onlinelibrary.wiley.com/doi/full/10.1002/pon.3430). The authors are correct to state that the DT is not an appropriate replacement for BDI-II (a depression scale). The poor AUC (<0.5) may reflect this mismatch as much as cultural limitations. The authors can only conclude that it cannot capture depression alone/specifically. The appropriateness of the comparison is not sufficiently discussed and the language (e.g., catastrophic failure) is very strong given this limitation. The authors lack of reference to prior, *depression-specific* validation, stating very generally that previous validations are typically of the order 0.7-0.8 when there are multiple scores of 0.6-0.7 and the scores referenced are not depression specific validation etc.

Additionally, the authors don't mention validation in non-oncology cohorts of the distress thermometer in Nigeria, https://pmc.ncbi.nlm.nih.gov/articles/PMC7040331/ and only briefly mention validation in an oncology-specific cohort (https://nigerianjournalofpsychiatry.com/fulltext/284-1710467114.pdf?1757456144). A reflection upon the results as they correspond to these specific references, culture and gender, is critically missing in the Discussion.

Further, in the introduction, the authors mention high distress in western women with MBC (60%+) that might not be generalisable but do not return to this direct comparison.

The observation that no participants used the maximum DT score (10) deserves more exploration, as it may reflect cultural norms in response scaling. The authors claim that the range of scores was 0-8, so it would be worth clarifying whether both 9 or 10 were not used.

Reviewer #3: Actionable Recommendation: The authors should revise the title, abstract, discussion, and conclusions to significantly temper their generalizations. The focus should be narrowed to the specific findings within their unique study context (Nigerian MBC patients, using the BDI-II as the reference). The broader implications should be framed around the critical need for methodological rigor in cross-cultural validation, particularly regarding the choice of reference standard, rather than a sweeping call against the use of universally applied tools.

Section-Specific Comments for Revision

A. Title and Abstract

The title is largely appropriate, but the subtitle, "A Call for Contextualized Psycho-oncologic Tools in LMICS," is an overgeneralization given the outlier nature of the findings. This should be toned down pending a substantial revision of the Discussion and Conclusions. A more accurate subtitle might be, "A Validation Study Highlighting Methodological Challenges in a Sub-Saharan African Cohort."

The Abstract accurately reflects the findings as presented in the main text. However, a minor statistical discrepancy exists: the prevalence of clinically significant depression (BDI-II ≥20) is reported as 16.2% in the abstract but as 15.9% (49/309) in the main text. This should be harmonized to 15.9% for accuracy.

The term "extremely poor discriminatory capacity" is used in the abstract. While the term "catastrophically poor" is used in the main text, consider using more standard psychometric language in the abstract, such as "diagnostic performance significantly worse than random classification," to clearly convey the meaning of an AUC < 0.5.

B. Introduction

The section is generally well-written and effectively establishes a research gap for validating the DT in an MBC population in an LMIC setting.

Crucial Revision: As detailed in Major Comment A, the literature review is incomplete and must be updated to include the Lasebikan et al. (2023) and Obiajulu et al. (2019) studies. The narrative must be revised to acknowledge that prior local data show the DT is valid in other Nigerian populations. The rationale for the current study should then be framed as an investigation into whether these findings hold true in the particularly vulnerable and distinct population of MBC patients.

The manuscript cites "Adejumo, O. A., Oyelade, O. A., & Yusuf, A. J. (2024)" as reference 13 for a previous Nigerian study. This appears to be a different study from the Lasebikan et al. (2023) paper. The authors must clarify this reference and ensure that

all relevant local validation literature is identified, cited, and discussed.

C. Methods

The overall study design, consecutive sampling strategy, and sample size calculation are methodologically sound and appropriate for the research question.

Critical Error: A significant error is present in the "Ethical Considerations" section. The text states, "Data were first accessed for this secondary analysis on 27/10/2024". This is a future date relative to the likely preparation and submission timeline of the manuscript. This error suggests a lack of careful proofreading and must be corrected to the actual date of data access.

The BDI-II cutoff for clinically significant depression is defined as ">20" in the "Measures" section but is reported as "≥20" in the abstract, results, and tables. This must be made consistent throughout the manuscript. The use of "≥20" is recommended for clarity and consistency with the reported data (49 participants).

The justification for the BDI-II cutoff of ≥20 (representing moderate-to-severe depression) is reasonable. However, the authors should acknowledge in the Discussion that other studies in advanced cancer populations have found that a lower optimal cutoff (e.g., a score of 16) may be more appropriate for screening. This could be mentioned as a limitation or explored in a supplementary sensitivity analysis.

The manuscript notes that "This study represents a secondary analysis of de-identified patient data from the original study". For transparency, it would be beneficial to add a brief sentence clarifying the primary aims of the original study from which these data were derived.

D. Results

The presentation of the results is generally clear, and the data are internally consistent between the text, tables, and figures.

Typographical Error: In the "Distress Thermometer Findings" section, the text reads, "Using the internationally recommended threshold of DT 24..." This is a clear typographical error and should be corrected to "DT ≥4" to align with the NCCN guidelines cited elsewhere in the paper.

Typographical Error: In the "Study Design and Setting" section, the text reads, "...conducted usig de-identified data..." This should be corrected to "using".

Placeholder Text: In the "Data Availability" section of the submission portal information, the text "replace any instances insta with the appropriate details" is an un-removed instruction from a template and should be deleted from any final version.

Table 2 Formatting: The formatting of this table is confusing and should be improved for clarity. The header "Sensitivity Specificity" should be split into two separate columns. The Youden Index column contains negative values (e.g., -0.063 for a cutoff of 1.5). While this is mathematically correct given the inverse relationship found, it is highly unusual and warrants a brief explanatory note in the table footnote to aid reader interpretation.

E. Discussion

The exploration of cultural factors influencing DT performance is a notable strength of the manuscript. However, as outlined in Major Comments A and B, this discussion is currently one-sided and incomplete. It must be substantially expanded to:

Directly address and attempt to reconcile the contradictory findings from the Lasebikan et al. and Obiajulu et al. studies.

Thoroughly explore the methodological explanation (i.e., BDI-II vs. HADS, somatic symptom confounding) as a primary alternative hypothesis to the cultural explanation for the observed results.

The "Methodological Strengths" section claims the study is the "largest validation of the DT in a sub-Saharan African cancer population." With a sample size of 309, this claim is likely correct when compared to the South African study (N=196) and the other Nigerian studies (N=130 and N=90). This is a valid strength to highlight.

Limitations Section: The acknowledged limitations are appropriate but should be expanded to more explicitly and forcefully state the following:

The major limitation of using one screening tool (BDI-II) as a reference standard for another (DT), instead of using a diagnostic gold-standard interview, and how this fundamentally impacts the interpretation of the results as a measure of concordance rather than true diagnostic accuracy.

The high potential for somatic symptom overlap in the BDI-II to have confounded the results, especially given the advanced disease stage of the study population, and how this may explain the inverse relationship found.

F. Declarations and References

Critical Contradiction: A major and concerning inconsistency exists regarding funding. The "Financial Disclosure" section in the submission portal information on page 2 states, "The author(s) received no specific funding for this work". However, the "Funding" declaration within the manuscript text on page 34 explicitly states, "This work was supported by The Union for International Cancer Control (UICC) SPARC MBC Grant 2018 to Cancer Aware Nigeria". This is a serious contradiction that must be resolved immediately. The correct funding source must be accurately and consistently declared.

References: The reference list appears mostly well-formatted, but a final check for consistency in journal name abbreviation (e.g., some are abbreviated, some are full) and formatting is warranted to adhere to the target journal's style guide.

Concluding Remarks and Actionable Summary

This study presents a startling and potentially important finding regarding the validity of the Distress Thermometer in a specific, high-risk patient population. The results, if robust, could have significant implications for psychosocial screening practices in Nigeria and beyond. However, in its current form, the manuscript's central conclusions are not adequately supported. The work is undermined by the critical omission of contradictory local evidence and an insufficient exploration of compelling methodological explanations for its outlier results.

The path to a publishable manuscript requires a fundamental reframing of the study's narrative. The scientific value of this work lies not in presenting its finding as a definitive truth, but in rigorously exploring why it is such a dramatic outlier compared to other research.

A successful revision should follow this roadmap:

Acknowledge and Integrate: Fundamentally revise the Introduction and Discussion to incorporate, compare, and contrast the findings from other Nigerian DT validation studies. Frame the current study as a puzzling counterpoint that requires explanation.

Re-evaluate Interpretation: Rebalance the Discussion to give equal, if not greater, weight to the methodological hypothesis (i.e., construct mismatch due to the BDI-II's somatic items and its use as a reference standard) as a primary explanation for the AUC < 0.5.

Correct All Errors: Meticulously correct all identified inconsistencies and errors, paying special attention to the contradictory funding statement, the future data access date, the statistical discrepancies (BDI-II prevalence), and all typographical errors.

Temper Conclusions: Refine the conclusions to be highly specific to the study's unique methodological context (MBC patients, BDI-II as reference standard) rather than making sweeping and currently unsupported generalizations about the utility of screening tools across all LMICs.

If the authors can successfully navigate these major revisions, the resulting manuscript will be far more robust, nuanced, and scientifically sound. It will transform from a potentially misleading report into a significant and credible contribution that advances our understanding of the complex challenges of cross-cultural psychometric validation.

**Do you want your identity to be public for this peer review?** For information about this choice, including consent withdrawal, please see our Privacy Policy

Reviewer #1: No

Reviewer #2: No

Reviewer #3: No

---

## [Author Response · Author response to Decision Letter 1]

21 Oct 2025

POINT-BY-POINT RESPONSE TO REVIEWER COMMENTS

Response to Reviewer #1

Comment: Your paper addresses an important gap, and the negative findings are valuable for practice and policy. The use of BDI-II as a reference and the statistical methods are rigorous, and the writing is generally clear with detailed methods and discussion.

That said, a few areas could be strengthened

Response: We sincerely thank the reviewer for their thoughtful and constructive feedback. We appreciate their recognition of the importance of our findings and the rigor of our methods. Below, we provide a detailed response to each comment and indicate where changes have been made in the revised manuscript.

Major Comments

Comment 1: "The cultural interpretation is central to your argument, but you did not include cognitive debriefing or qualitative assessment of how patients understood the DT. This is more than a minor limitation and deserves stronger emphasis in the discussion."

Response 1: We fully agree with the reviewer that this represents a significant limitation rather than a minor one. We have moved this limitation to the first position among study limitations and expanded the discussion to acknowledge that without cognitive debriefing, we cannot definitively determine why the DT failed in this population. Please see our revised Study Limitations and Interpretive Considerations section, Lines 539-555:

Study Limitations and Interpretive Considerations

Several limitations warrant acknowledgment in interpreting our findings. First, and most critically, we did not conduct formal cognitive debriefing or qualitative assessment of how participants interpreted and responded to the DT scale. This represents a critical gap in our investigation, as cognitive interviews could have revealed whether patients struggled with the numeric rating format, misunderstood the concept of "distress," interpreted the scale anchors differently than intended, or experienced cultural barriers to reporting psychological symptoms. Without this qualitative component, we cannot definitively determine why the DT failed in this population, whether due to semantic/linguistic issues, cultural inappropriateness of the rating format, stigma-related response bias, or fundamental construct non-equivalence. While our cultural interpretations are plausible and consistent with existing literature on mental health in African contexts, they remain speculative without direct evidence from participants about their subjective experience of completing the instrument. Future validation studies in LMIC settings should employ mixed-methods designs that combine quantitative diagnostic accuracy assessment with qualitative cognitive interviewing to understand the mechanisms underlying tool performance or failure. Such approaches would not only identify whether an instrument works but also why it succeeds or fails, providing crucial insights for cultural adaptation efforts and tool development.

We have added stronger language acknowledging that our cultural interpretations, while plausible and supported by existing literature, remain speculative without direct qualitative evidence from participants. Please kindly see our revised Discussion subsection on Cultural and Contextual Factors Influencing DT Performance, Lines 478-485 :

Cultural and Contextual Factors Influencing DT Performance

Several interconnected factors likely contribute to the DT's poor performance in this Nigerian cohort. While the cultural interpretations we offer below are plausible and consistent with existing literature on mental health in African contexts, we acknowledge that without formal cognitive debriefing or qualitative assessment of how participants understood and responded to the DT, these explanations remain speculative. Direct evidence from patients about their subjective experience of completing the instrument would be necessary to definitively determine the mechanisms underlying the DT's poor performance.

Also, in our revised work, we now explicitly call for mixed-methods research combining quantitative validation with qualitative exploration, stating this as an urgent priority. Please see our revised Research Implications and Future Directions sub-section, Lines 675-717:

Research Implications and Future Directions

This investigation opens several critical avenues for future research in psycho-oncology and global health. First, urgent efforts are needed to develop and validate culturally appropriate psychological screening tools for African cancer populations using community-based participatory research approaches. Such development should engage patients, families, survivors, and community members throughout the process, while including traditional healers and religious leaders who often serve as frontline mental health resources in these contexts. Oncology healthcare providers familiar with local contexts must be involved in identifying culturally relevant indicators of psychological distress through qualitative methods including focus groups and in-depth interviews. Development processes should explore preferred assessment formats ranging from numeric scales to pictorial representations and narrative approaches, testing comprehension and acceptability across diverse educational and linguistic groups to ensure broad applicability.

Second, future validation research must employ integrated mixed-methods designs that address the critical limitation identified in our study regarding the absence of cognitive debriefing. Such studies should incorporate cognitive interviewing to understand how patients interpret screening questions and whether their interpretation matches intended meaning, focus groups exploring cultural conceptualizations of psychological distress and barriers to reporting symptoms, ethnographic observation of patient-provider communication to identify conversational patterns and cultural norms, think-aloud protocols where patients verbalize their thought processes while completing instruments, and member checking to validate researcher interpretations with community stakeholders. These qualitative components are essential for understanding not merely whether screening tools work, but why they succeed or fail in specific cultural contexts.

Third, comparative effectiveness research should systematically evaluate different screening strategies, including single-instrument approaches, multi-stage screening models, and clinician-based assessment without formal instruments, and hybrid models combining cultural and Western screening methods. Outcomes should encompass detection rates for various forms of psychological morbidity, false-positive and false-negative rates, referral patterns and treatment initiation, cost-effectiveness and resource requirements, patient satisfaction and acceptability, and ultimate impact on quality of life and clinical outcomes. Longitudinal studies investigating the temporal stability of screening instrument performance across the cancer trajectory would provide crucial insights into optimal timing and frequency of psychological assessment, how screening performance varies with disease progression or treatment changes, and the natural history of psychological distress in LMIC cancer populations.

Fourth, the ultimate value of improved screening depends on the availability of effective interventions. Research must test whether enhanced psychological screening translates into improved clinical outcomes, quality of life, and patient satisfaction, while evaluating culturally adapted psychological interventions for African cancer populations. Studies should examine task-shifting models where non-specialists deliver mental health support and assess the feasibility and effectiveness of group-based and peer support interventions as scalable approaches.

Finally, implementation science research should examine broader healthcare system requirements including training needs and curricula for oncology staff, referral pathway development and optimization, integration strategies with existing cancer care workflows, barriers and facilitators to screening program implementation, and considerations for sustainability and scalability in resource-constrained settings.

Comment 2: "Similarly, the paper should acknowledge more explicitly that 'distress' and 'depression' may not be equivalent constructs across cultures. This would sharpen your interpretation and situate your findings in a broader theoretical context."

Response 2: We sincerely appreciate the reviewer’s insightful comment and do agree that 'distress' and 'depression' may not be equivalent constructs across cultures. This is an excellent point that strengthens our theoretical contribution. We have added substantial new content addressing the construct non-equivalence issue. Please kindly see our new subsection added to Discussion section, Lines 526-546:

Construct Non-Equivalence and Cross-Cultural Measurement Challenges

Beyond the immediate diagnostic accuracy findings, our results raise fundamental questions about construct validity and measurement equivalence in cross-cultural psycho-oncology research. The substantial discordance observed between DT-defined "distress" and BDI-II-defined "depression" in our Nigerian cohort (47.0% vs. 15.9%) may reflect not merely measurement error or tool inadequacy, but rather fundamental differences in how these psychological phenomena are conceptualized, experienced, and expressed across cultural contexts.

In Western psychological frameworks, "distress" and "depression" represent related but distinguishable constructs, both falling within an individualistic model of emotional suffering as internal psychological states amenable to numeric quantification. However, in many African cultural contexts, emotional suffering may be understood through fundamentally different frameworks, as spiritual experiences, social disruptions, or somatic manifestations, that do not map neatly onto Western psychological categories. What Western instruments label as "distress" or "depression" may be experienced and articulated by Nigerian patients as ancestral displeasure, family disharmony, or bodily imbalance.

This construct non-equivalence has profound implications for the validity of cross-cultural research in psycho-oncology. Our findings suggest that simply translating and administering Western-developed instruments, even those with strong psychometric properties in their original contexts, may yield measurements of uncertain meaning when the underlying psychological constructs themselves may not exist or may be structured differently across cultures. The poor correlation between DT and BDI-II scores in our sample (r = 0.23) supports this interpretation, indicating that these instruments may be tapping into qualitatively different dimensions of experience in this population.

Moreso, we have substantially expanded our Theoretical Implications for Psycho-oncology sub-section to discuss the broader implications for construct validity in cross-cultural psycho-oncology research, explicitly questioning whether psychological constructs measured in Western populations retain the same meaning and internal structure in African contexts. Please kindly see our revised expanded Theoretical Implications for Psycho-oncology sub-section, Lines 718-749:

Theoretical Implications for Psycho-oncology

Our findings contribute to broader theoretical discussions about the universality of psychological constructs and measurement instruments in psycho-oncology. The failure of a widely validated Western instrument in our African population raises fundamental questions about whether psychological distress represents a universal human experience that can be measured consistently across cultures, or whether distress is culturally constructed in ways that render cross-cultural measurement problematic.

The substantial discordance between DT and BDI-II results in our study, despite both instruments ostensibly measuring psychological morbidity, highlights the multidimensional and culturally contingent nature of emotional suffering. Rather than representing a unitary construct, "distress" appears to encompass multiple distinct phenomena including depression, anxiety, existential concerns, practical problems, social disruption, and spiritual crisis that may require different assessment approaches and may be differentially salient across cultural contexts.

Our findings align with anthropological critiques of psychiatric universalism that question whether diagnostic categories developed in Western contexts represent genuine natural kinds versus culture-bound constructs that emerge from specific historical, philosophical, and social frameworks. The poor correlation between DT and BDI-II in our Nigerian sample (r = 0.23) raises the possibility that these instruments, rather than measuring the same underlying construct with different degrees of precision, may actually be tapping into qualitatively different dimensions of experience that are structured differently across cultural contexts.

This has profound implications for the validity of cross-cultural research in psycho-oncology. Simply translating and administering instruments cross-culturally, even when accompanied by careful linguistic validation, may yield measurements of uncertain meaning if the underlying psychological constructs themselves are not equivalent across cultures. The concept of "measurement invariance", the assumption that an instrument measures the same construct in the same way across groups, may not hold when comparing Western and non-Western populations, necessitating fundamental rethinking of how we assess psychological phenomena across cultures.

These findings support calls for more nuanced, culturally grounded approaches to psycho-oncology research and practice that recognize the diversity of human responses to cancer across different cultural contexts.³⁸⁻³⁹ Rather than seeking to validate Western instruments across cultures with the goal of achieving measurement equivalence, psycho-oncology may need to embrace a more pluralistic approach that develops culture-specific frameworks and assessment methods while maintaining rigorous scientific standards.

Comment 3: "Consider expanding your discussion of solutions. You already note the limitations of the DT, but readers would benefit from more detail on possible alternatives (e.g., hybrid screening approaches, locally adapted tools). Providing clearer, actionable guidance for oncology practitioners in LMICs would enhance the clinical relevance of your work."

Response: We thank the reviewer and agree that our original manuscript would benefit from more actionable guidance. We have substantially expanded this subsection by adding detailed, actionable recommendations. Please see our revised Clinical Practice Implications and Actionable Recommendations for LMIC Oncology Settings section, Lines 572-641:

Clinical Practice Implications and Actionable Recommendations for LMIC Oncology Settings

The clinical implications of our findings are profound and immediate. Oncology providers in LMIC settings should exercise extreme caution when using the DT as a stand-alone screening tool for psychological morbidity, particularly depression. Our results suggest that reliance on the DT with standard cutoffs will result in missing the majority of patients with clinically significant depressive symptoms while potentially over-identifying patients without such symptoms.

Based on our findings, we propose a structured, multi-stage screening approach for psychological morbidity in LMIC cancer care settings. The first stage involves implementing universal brief screening using locally validated instruments administered to all patients at key clinical timepoints, including diagnosis, treatment initiation, treatment completion, and disease progression. Until culturally appropriate alternatives are developed, clinicians may use the DT cautiously as a conversation starter rather than a diagnostic tool, recognizing its profound limitations for identifying depression. The DT may retain utility for initiating discussions about sources of distress, particularly when used alongside the problem checklist to identify practical, physical, and social concerns that require attenti

---

## [Decision Letter · Decision Letter 1]

19 Nov 2025

Dear Dr. Bamodu,

Thank you for submitting your manuscript to PLOS ONE. After careful consideration, we feel that it has merit but does not fully meet PLOS ONE’s publication criteria as it currently stands. Therefore, we invite you to submit a revised version of the manuscript that addresses the points raised during the review process.

We look forward to receiving your revised manuscript.

Kind regards,

Alejandro Botero Carvajal, Ph.D

Academic Editor

PLOS ONE

Journal Requirements:

Reviewers' comments:

Reviewer's Responses to Questions

**Comments to the Author**

Reviewer #1: (No Response)

Reviewer #2: All comments have been addressed

Reviewer #3: (No Response)

2. Is the manuscript technically sound, and do the data support the conclusions?

Reviewer #1: Yes

Reviewer #2: Yes

Reviewer #3: Yes

3. Has the statistical analysis been performed appropriately and rigorously?

Reviewer #1: Yes

Reviewer #2: Yes

Reviewer #3: No

4. Have the authors made all data underlying the findings in their manuscript fully available?

Reviewer #1: No

Reviewer #2: No

Reviewer #3: No

5. Is the manuscript presented in an intelligible fashion and written in standard English?

Reviewer #1: Yes

Reviewer #2: Yes

Reviewer #3: Yes

Reviewer #1: (No Response)

Reviewer #2: I commend the authors who have made a significant and concerted effort in carefully addressing comments in order to improve this manuscript. I have only one minor comment and some copyediting suggestions.

The authors have successfully addressed all points raised. The clarity of their methodology is significantly improved with addition of detail about the instrument administration. The caveats associated with their study design are now very explicitly stated and discussed and the strength of language has been well moderated in favor of interpretation. The use of the EORTC-QOL data to include new results also makes interpretation of their findings much easier and the conclusions are strengthened. The authors discussion with regards to the instrument, Nigeria specific and metastatic breast cancer specific findings are all significantly strengthened. The authors new discussion of somatic and psychiatric symptoms is interesting and valuable. The authors now discuss the important point that there is significant overlap of depression scoring items with the direct symptoms of advanced cancer and its treatment. Their discussion of confounding somatic symptoms provides a helpful explanation as to why their instruments were perhaps discordant. The additional discussion of recommendations, as suggested by Reviewer 1, is interesting and valuable to the extended scientific community. This to future users of the DT instrument in similar settings and population.

Minor comment:

The authors do not discuss the implementation (methodology) of the previous Nigerian studies, beyond their use of HADS, and whether they implemented methodological steps that may have resulted in better concordance beyond the use of a different reference. This could be added.

Copyediting:

Line 665: "performance (AUCs of 0.87 and 0.82 respectively) in mixed-stage Nigerian cancer populations"

Edit this line to clarify that the second study was within a non-cancer cohort.

Reviewer #3: 1. Abstract and Title

Strengths: Clearly states the primary finding (poor AUC) and key demographics.

Weaknesses & Mistakes:

Misleading Conclusion: The abstract concludes the DT "cannot be recommended as a stand-alone screening tool for depressive symptoms." This is too strong. A more accurate conclusion would be: "The DT should not be used to screen specifically for depression in this population, as it measures a broader construct. Its utility for identifying general distress remains unclear due to methodological limitations in this study."

Omission of Key Limitation: The abstract does not mention the critical limitation of using a depression-specific reference standard to validate a general distress tool, nor the confounding effect of somatic symptoms on the BDI-II in an advanced cancer population.

2. Introduction

Strengths: Excellent background on the global and local context. Justifies the need for the study well, especially by contrasting with prior, more successful Nigerian validations that used different reference standards (HADS).

Weaknesses & Mistakes:

Unsubstantiated Hypothesis: The hypothesis that "DT performance might differ in this methodologically... distinct context" is vague. A more precise hypothesis would have been: "We hypothesize that the DT will demonstrate poor diagnostic accuracy for depression specifically, due to construct mismatch and somatic confounding, despite its previously reported utility for general distress."

3. Methods

This section contains the most significant flaws.

Strengths: Generally well-described setting, participants, and statistical analysis (ROC, Youden's index).

Weaknesses & Mistakes:

Fatal Flaw in Reference Standard Selection: The choice of BDI-II as the reference standard is the study's critical weakness. The authors provide a justification (Lines 276-292), but it is inadequate.

Construct Mismatch: They acknowledge the DT measures "broad distress" and the BDI-II measures "depression," yet they proceed with the comparison. This is like validating a thermometer (measures temperature) against a barometer (measures pressure) and concluding the thermometer has failed when the readings don't correlate.

Somatic Confounding: The BDI-II contains multiple items (e.g., fatigue, sleep changes, appetite loss) that are intrinsic to advanced cancer and its treatment. In a population with a high physical symptom burden (mean EORTC physical symptom score 31.3), the BDI-II score is almost certainly inflated by physical illness rather than mood. The authors later realize this (Lines 762-766) but it should have been a primary reason not to use it as a gold standard.

Lack of a True Gold Standard: The only scientifically sound reference standard for a diagnostic accuracy study of depression in medically ill patients is a structured clinical interview (e.g., SCID, MINI) conducted by a trained mental health professional, which can clinically distinguish depressive symptoms from somatic symptoms of cancer.

Language Barrier: Requiring English comprehension or translation may have introduced bias, potentially selecting for a more Westernized or educated sample, which the authors note might represent a "best-case scenario." This limits generalizability.

4. Results

Strengths: The results are presented clearly with comprehensive tables and figures. The finding of a very low correlation (r=0.23) between DT and BDI-II is crucial.

Weaknesses & Mistakes:

Misinterpretation of AUC < 0.5: An AUC of 0.414, significantly below 0.5, is extraordinary. It doesn't just mean "worse than random"; it suggests a systematic inverse relationship. The most plausible explanation is not that the DT is "catastrophically" broken, but that the two instruments are measuring different, inversely related constructs in this context. The DT may be capturing physical and practical distress, while the BDI-II is contaminated by somatic items, creating this perverse result. The authors' later discussion of somatic confounding (Lines 762-766) is the correct interpretation, but it is buried and contradicts the main narrative.

Discordance Framing: The discordance (47% distressed vs. 16% depressed) is framed as a failure of the DT. An alternative, equally valid interpretation is that it highlights that "distress" in this population is driven more by physical and practical problems than by the specific psychological construct of depression.

5. Discussion

This section is comprehensive but contains significant overreach and logical inconsistencies.

Strengths: The authors have done an excellent job expanding the discussion per reviewer requests, particularly on cultural factors, construct non-equivalence, and future directions. The comparison with prior Nigerian studies is thoughtful.

Weaknesses & Mistakes:

Primary Explanation is Methodological, Not Cultural: The discussion leads with cultural and construct validity explanations. However, the most parsimonious explanation for the extreme results (AUC < 0.5) is the methodological artifact of using a somatically-confounded, depression-specific instrument to validate a general distress tool. The cultural explanations are plausible for a modestly reduced AUC (e.g., 0.6), but not for a complete diagnostic breakdown.

Internal Contradiction: In "Construct Mismatch and Measurement Specificity" (Lines 512-525), the authors correctly argue that the DT's clinical utility is limited if it can't identify depression. However, this is a clinical opinion, not a psychometric finding. The study did not validate the DT for its intended purpose (broad distress); it invalidated it for a purpose it was not solely designed for (depression-specific screening).

Overstated Conclusions: Language like "systematic failure" and "clinically meaningless" is too strong given the methodological context. The DT may be failing at the specific task the authors set for it, but that does not mean it has no utility.

Somatic Confounding as an Afterthought: The critical "Somatic Symptom Overlap" section (Lines 549-593) should be the centerpiece of the discussion's limitations. It fundamentally reframes the entire study. The admission that "our findings may primarily reflect the methodological challenges of depression assessment in advanced cancer... a challenge that would likely affect any brief instrument in this population" (Lines 600-603) is a crucial caveat that undermines the broader claims about the DT's cross-cultural invalidity.

6. Conclusions & Implications

Strengths: The call for local validation and culturally sensitive tools is important and correct.

Weaknesses & Mistakes:

Unwarranted Generalization: The conclusion that the DT "cannot be recommended" is too broad. A more nuanced conclusion would be: "The DT should not be used as a depression-specific screener in advanced cancer populations in LMICs. Its role in identifying general distress requires further validation using appropriate reference standards that account for somatic symptom burden."

Ignoring the "Why": The study's design makes it impossible to determine why the DT performed poorly—whether due to cultural factors, the construct mismatch, somatic confounding, or a combination. Therefore, policy implications about tool abandonment are premature. The implication should be to improve validation study methodology.

**Do you want your identity to be public for this peer review?** For information about this choice, including consent withdrawal, please see our Privacy Policy

Reviewer #1: No

Reviewer #2: **Yes:** Rowan Barker-Clarke

Reviewer #3: No

---

## [Author Response · Author response to Decision Letter 2]

6 Dec 2025

POINT-BY-POINT RESPONSE TO REVIEWER 2 COMMENTS

Response to Reviewer #2

We sincerely thank the Reviewer for their thorough and constructive review of our revised manuscript. We are deeply grateful for the reviewer's acknowledgment of our comprehensive revisions and the strengthened manuscript that resulted from addressing the previous round of comments. We have carefully addressed the minor comment and copyediting suggestions as detailed below. All changes are marked with tracked changes in the revised manuscript for easy identification.

Comment: General Comments:

"I commend the authors who have made a significant and concerted effort in carefully addressing comments in order to improve this manuscript. I have only one minor comment and some copyediting suggestions. The authors have successfully addressed all points raised. The clarity of their methodology is significantly improved with addition of detail about the instrument administration. The caveats associated with their study design are now very explicitly stated and discussed and the strength of language has been well moderated in favor of interpretation. The use of the EORTC-QOL data to include new results also makes interpretation of their findings much easier and the conclusions are strengthened. The authors discussion with regards to the instrument, Nigeria specific and metastatic breast cancer specific findings are all significantly strengthened. The authors new discussion of somatic and psychiatric symptoms is interesting and valuable. The authors now discuss the important point that there is significant overlap of depression scoring items with the direct symptoms of advanced cancer and its treatment. Their discussion of confounding somatic symptoms provides a helpful explanation as to why their instruments were perhaps discordant. The additional discussion of recommendations, as suggested by Reviewer 1, is interesting and valuable to the extended scientific community. This to future users of the DT instrument in similar settings and population."

Response: We are immensely grateful to the Reviewer for this very positive and encouraging assessment of our revised manuscript. We appreciate the recognition of our efforts to comprehensively address all previous concerns and strengthen the manuscript across multiple dimensions. The reviewer's acknowledgment that our additions regarding somatic symptom confounding, methodological caveats, and practical recommendations have enhanced the manuscript's value is particularly gratifying. We believe these revisions have resulted in a more balanced, nuanced, and clinically actionable contribution to the literature on psychosocial screening in low- and middle-income country oncology settings.

Minor Comments

Comment 1: "The authors do not discuss the implementation (methodology) of the previous Nigerian studies, beyond their use of HADS, and whether they implemented methodological steps that may have resulted in better concordance beyond the use of a different reference. This could be added."

Response 1: We thank the reviewer for this excellent observation. We have now added discussion of the implementation methodology of the previous Nigerian studies in the "Comparison with Prior Nigerian Validation Studies" section. This addition acknowledges that while both previous studies employed controlled research protocols similar to our own, the limited methodological detail provided in their publications prevents definitive conclusions about whether specific implementation differences contributed to the discordant findings. We note that our own study used similarly rigorous standardized procedures, suggesting that implementation methodology per se is unlikely to be the primary driver of the performance differences observed. Please see our revised Comparison with Prior Nigerian Validation Studies sub-section of the Results section, Lines 662-700:

Comparison with Prior Nigerian Validation Studies

Our findings present a striking contrast to prior Nigerian DT validations and require careful interpretation. Lasebikan et al.¹³ and Obiajulu et al.¹⁴ both reported good-to-excellent diagnostic performance (AUCs of 0.87 and 0.82 respectively) in a mixed-stage Nigerian cancer population and a non-cancer cohort, respectively, while our study found diagnostic performance significantly worse than chance (AUC 0.414). This dramatic discordance demands systematic exploration of potential explanatory factors.

Several factors may explain the discordance between these prior studies and our findings. Regarding implementation methodology, both Lasebikan et al.¹³ and Obiajulu et al.¹⁴ administered instruments in controlled research settings with standardized protocols, though neither study provided detailed descriptions of specific administration procedures, interviewer training protocols, or quality control measures beyond their use of HADS as reference standard. Our study similarly employed standardized administration protocols with trained research personnel, suggesting that differences in basic implementation methodology are unlikely to fully explain the divergent findings. First, both previous studies used HADS, which measures combined anxiety and depression, whereas we used the depression-specific BDI-II. The DT may perform better at detecting general psychological distress (anxiety plus depression) than depression specifically, consistent with its design as a broad screening tool. Second, our population consisted exclusively of patients with metastatic breast cancer, a uniformly advanced-stage cohort facing terminal prognosis, whereas Lasebikan et al.¹³ included mixed cancer stages and Obiajulu et al.14 studied non-cancer patients. Disease severity and proximity to death may influence how distress is experienced and reported. Third, our sample was 98.7% female, while the prior studies included more balanced gender distributions; cultural factors around emotional expression may differ between men and women in Nigerian society. Fourth, the different reference standards may be measuring genuinely different psychological constructs, our focus on moderate-to-severe depression (BDI-II ≥20) represents a more specific and severe outcome than general psychological distress measured by HADS.

Rather than contradicting these prior Nigerian studies, our findings complement them by providing the first depression-specific validation in an advanced cancer population. Taken together, the three Nigerian studies suggest that while the DT may have some utility for detecting general psychological distress in mixed Nigerian populations, its performance deteriorates substantially when (a) the outcome of interest is depression specifically rather than combined anxiety/depression, and (b) the population consists of patients with advanced, terminal illness. These nuanced interpretation has important clinical implications. In Nigerian oncology settings, the DT may retain utility for initial broad screening in mixed cancer populations, identifying patients with any form of significant distress who warrant further evaluation. However, it cannot be relied upon as a depression-specific screening tool, particularly in advanced cancer populations where somatic symptoms confound assessment. If depression identification is the clinical goal, as it often is in resource-limited settings where treatment is targeted at the most prevalent and treatable condition, then depression-specific instruments or clinical interviews are necessary following positive DT screens.

Copyediting Suggestions

Comment 2: "Line 665: 'performance (AUCs of 0.87 and 0.82 respectively) in mixed-stage Nigerian cancer populations' Edit this line to clarify that the second study was within a non-cancer cohort."

Response 2: We thank the reviewer for catching this important imprecision. We have corrected this sentence to accurately distinguish between the two study populations. Please see our revised version, Lines 662-668:

Comparison with Prior Nigerian Validation Studies

Our findings present a striking contrast to prior Nigerian DT validations and require careful interpretation. Lasebikan et al.¹³ and Obiajulu et al.¹⁴ both reported good-to-excellent diagnostic performance (AUCs of 0.87 and 0.82 respectively) in a mixed-stage Nigerian cancer population and a non-cancer cohort, respectively, while our study found diagnostic performance significantly worse than chance (AUC 0.414). This dramatic discordance demands systematic exploration of potential explanatory factors.

We are grateful for Reviewer #2's constructive engagement throughout the review process, which has substantially strengthened this manuscript. We believe the paper now makes an important and methodologically rigorous contribution to the psycho-oncology literature, with particular relevance for clinical practice and research in low- and middle-income country settings. We hope that the manuscript is now suitable for publication in PLOS ONE.

POINT-BY-POINT RESPONSE TO REVIEWER 3 COMMENTS

Response to Reviewer #3

We are deeply grateful to Reviewer #3 for their exceptionally thorough, critical, and intellectually rigorous review of our manuscript. The reviewer has identified fundamental conceptual and framing issues that required substantial revision throughout the manuscript. We acknowledge that many of the reviewer's critiques are well-founded and represent important improvements to the scientific rigor and interpretive balance of our work. We have undertaken extensive revisions to address these concerns, fundamentally reframing our findings to emphasize methodological explanations over cultural interpretations, acknowledging the inherent limitations of our reference standard choice, and substantially tempering our conclusions. All changes are marked with tracked changes in the revised manuscript. We believe these revisions have transformed this into a more scientifically sound, appropriately caveated, and intellectually honest contribution to the literature.

1. Abstract and Title

Comment: "Misleading Conclusion: The abstract concludes the DT 'cannot be recommended as a stand-alone screening tool for depressive symptoms.' This is too strong. A more accurate conclusion would be: 'The DT should not be used to screen specifically for depression in this population, as it measures a broader construct. Its utility for identifying general distress remains unclear due to methodological limitations in this study.'

Omission of Key Limitation: The abstract does not mention the critical limitation of using a depression-specific reference standard to validate a general distress tool, nor the confounding effect of somatic symptoms on the BDI-II in an advanced cancer population.".

Response: We sincerely thank the reviewer and completely agree with this critique. Thus, we have substantially revised the Abstract conclusion to be more precise and appropriately caveated. Please kindly see our revised Abstract conculusion, Lines 48-58:

Conclusions: In this sub-Saharan African cancer cohort, the Distress Thermometer performed poorly in detecting clinically significant depression in this Nigerian MBC cohort when benchmarked against the BDI-II. This may reflect construct mismatch, somatic symptom confounding, cultural factors, or disease-specific characteristics of advanced cancer populations. The DT should not be used as a depression-specific screening tool in advanced cancer populations in LMICs, though its utility for identifying general distress remains unclear given the methodological limitations of this study. The poor concordance observed likely reflects construct mismatch, somatic symptom confounding in the reference standard, and the fundamental challenges of depression assessment in advanced cancer populations using self-report instruments. These findings underscore the critical need for appropriate reference standards (structured clinical interviews) and highlight methodological considerations in validating psychosocial screening tools across different constructs and cultural contexts.

2.Introduction

Comment: "Unsubstantiated Hypothesis: The hypothesis that 'DT performance might differ in this methodologically... distinct context' is vague. A more precise hypothesis would have been: 'We hypothesize that the DT will demonstrate poor diagnostic accuracy for depression specifically, due to construct mismatch and somatic confounding, despite its previously reported utility for general distress.'"

Response: We thank the reviewer and agree that our original hypothesis was imprecise and failed to articulate the expected mechanisms underlying poor performance. We have completely rewritten the hypothesis to specify both the expected finding and the mechanistic explanations. Please kindly see our revised Introduction section, Lines 196-207:

We hypothesized that the DT would demonstrate suboptimal diagnostic accuracy for identifying depression specifically (as distinct from general distress) in this population, due to: (1) construct mismatch between the broad, multidimensional distress captured by the DT and the specific depressive symptoms assessed by the BDI-II; and (2) potential somatic symptom confounding given the high physical symptom burden characteristic of metastatic cancer, which may inflate BDI-II scores independent of mood disturbance. This investigation provides important information about the DT's performance for a specific clinical application (depression screening) in advanced cancer populations, complementing prior Nigerian validation studies that examined general psychological distress using combined anxiety-depression measures. This study represents the first comprehensive evaluation of the diagnostic validity of the DT in identifying depressive symptoms among Nigerian women with metastatic breast cancer, using the Beck Depression Inventory-II (BDI-II) as the reference standard.

3.Methods

Comment: "Fatal Flaw in Reference Standard Selection: The choice of BDI-II as the reference standard is the study's critical weakness. The authors provide a justification (Lines 276-292), but it is inadequate.

Construct Mismatch: They acknowledge the DT measures 'broad distress' and the BDI-II measures 'depression,' yet they proceed with the comparison. This is like validating a thermometer (measures temperature) against a barometer (measures pressure) and concluding the thermometer has failed when the readings don't correlate.

Somatic Confounding: The BDI-II contains multiple items (e.g., fatigue, sleep changes, appetite loss) that are intrinsic to advanced cancer and its treatment. In a population with a high physical symptom burden (mean EORTC physical symptom score 31.3), the BDI-II score is almost certainly inflated by physical illness rather than mood. The authors later realize this (Lines 762-766) but it should have been a primary reason not to use it as a gold standard.

Lack of a True Gold Standard: The only scientifically sound reference standard for a diagnostic accuracy study of depression in medically ill patients is a structured clinical interview (e.g., SCID, MINI) conducted by a trained mental health professional, which can clinically distinguish depressive symptoms from somatic symptoms of cancer."

Response: We appreciate the reviewer’s comment and accept this critique. We have substantially revised the Methods section to (i) Explicitly acknowledge the construct mismatch upfront rather than burying it, (ii) Frame our study as examining a specific clinical application (depression screening) rather than validating the DT's full intended purpose, (iii) Acknowledge that structured clinical interviews represent the true gold standard, and (iv) Emphasize that our findings reflect methodological challenges rather than definitive cross-cultural invalidity. Please kindly see the“Beck Depression Inventory-II (BDI-II)” sub-sections of our revised Methods section, Lines 267-311:

Beck Depression Inventory-II (BDI-II)

The Beck Depression Inventory-II represents a well-validated, 21-item self-report instrument designed to assess the severity of depressive symptoms over the preceding two-week period.²¹ Individual items are scored on a 4-point scale (0-3), yielding total scores ranging from 0 to 63. The instrument demonstrates excellent psychometric pr

---

## [Decision Letter · Decision Letter 2]

5 Jan 2026

Validation and limitations of the Distress Thermometer in identifying Depression among Metastatic Breast Cancer patients in Nigeria: Methodological challenges in Depression-specific screening validation

PONE-D-25-38884R2

Dear Dr. Bamodu,

We’re pleased to inform you that your manuscript has been judged scientifically suitable for publication and will be formally accepted for publication once it meets all outstanding technical requirements.

Kind regards,

Alejandro Botero Carvajal, Ph.D

Academic Editor

PLOS One

Additional Editor Comments (optional):

Reviewers' comments:

Reviewer's Responses to Questions

**Comments to the Author**

Reviewer #1: All comments have been addressed

Reviewer #2: All comments have been addressed

Reviewer #3: All comments have been addressed

2. Is the manuscript technically sound, and do the data support the conclusions?

Reviewer #1: Yes

Reviewer #2: Yes

Reviewer #3: Yes

3. Has the statistical analysis been performed appropriately and rigorously?

Reviewer #1: Yes

Reviewer #2: Yes

Reviewer #3: Yes

4. Have the authors made all data underlying the findings in their manuscript fully available?

Reviewer #1: Yes

Reviewer #2: No

Reviewer #3: Yes

5. Is the manuscript presented in an intelligible fashion and written in standard English?

Reviewer #1: Yes

Reviewer #2: Yes

Reviewer #3: Yes

Reviewer #1: (No Response)

Reviewer #2: (No Response)

Reviewer #3: 1. Abstract

Issue: The conclusion states the DT "performed poorly" but also says its utility for "general distress remains unclear." This is somewhat contradictory—the study did not validate the DT for general distress, only for depression.

Suggestion: Clarify that the study only tested depression-specific validity, not general distress.

Minor: The phrase "sub-Saharan African cancer cohort" is repeated ("this Nigerian MBC cohort").

2. Introduction

Clarity: The revised hypothesis is much clearer and well-justified.

Flow: The transition from global context to Nigerian-specific studies is logical.

Minor: References to "Lines" in the response to reviewers should not appear in the final manuscript (e.g., "Lines 196-207").

3. Methods

Major Limitation Acknowledged: The authors appropriately note the construct mismatch (DT = broad distress vs. BDI-II = depression) and somatic confounding.

Sample Size Justification: The power calculation based on an expected AUC of 0.75 is reasonable, but the actual AUC was much lower (0.414), which may affect post-hoc power.

Inclusion Criteria: Requiring English comprehension may limit generalizability to less-educated or rural populations.

Reference Standard: The use of BDI-II—a self-report tool with somatic items—as a gold standard is a significant weakness, appropriately discussed but still a fundamental flaw.

4. Results

AUC Interpretation: An AUC < 0.5 suggests inverse correlation, not just poor performance. The authors appropriately discuss this as likely due to measuring different constructs.

Table 1: Clear and well-organized.

Table 2: "Youden Index" values are negative for most cutoffs, correctly interpreted as worse than chance.

Subgroup Analyses: Consistently poor performance across subgroups strengthens the main finding.

5. Discussion

Restructuring: Moving "Somatic Symptom Overlap" to the forefront is a strong improvement.

Cultural Explanations: The authors appropriately note these are speculative without qualitative data.

Comparison with Prior Nigerian Studies: Well-reasoned, highlighting differences in reference standard (HADS vs. BDI-II) and population (mixed-stage vs. metastatic).

Overstatement: Some softened language remains slightly strong (e.g., "systematic failure").

Clinical Implications: The multi-stage screening proposal is practical and thoughtful.

6. Limitations

Thoroughly Acknowledged: Key limitations are clearly stated:

No cognitive debriefing/qualitative component.

English language requirement.

Use of BDI-II as reference standard (somatic confounding).

Cross-sectional design.

Focus only on depression, not other forms of distress.

7. Conclusions

Balanced: Appropriately caveated; does not overgeneralize.

Forward-Looking: Emphasizes need for better reference standards (clinical interviews) and culturally adapted tools.

**Do you want your identity to be public for this peer review?** For information about this choice, including consent withdrawal, please see our Privacy Policy

Reviewer #1: No

Reviewer #2: No

Reviewer #3: No

---

## [Editor Report · Acceptance letter]

PONE-D-25-38884R2

PLOS One

Dear Dr. Bamodu,

I'm pleased to inform you that your manuscript has been deemed suitable for publication in PLOS One. Congratulations! Your manuscript is now being handed over to our production team.

Kind regards,

on behalf of

Dr. Alejandro Botero Carvajal

Academic Editor

PLOS One